# Marich: A Query-efficient Distributionally Equivalent Model Extraction Attack using Public Data

**Pratik Karmakar**[*]
School of Computing, National University of Singapore
CNRS@CREATE Ltd, 1 Create Way, Singapore
`pratik.karmakar@u.nus.edu`

**Debabrota Basu**
Équipe Scool, Univ. Lille, Inria, CNRS, Centrale Lille
UMR 9189- CRIStAL, F-59000 Lille, France
`debabrota.basu@inria.fr`

## Abstract

We study design of black-box model extraction attacks that can *send minimal number of queries from* a *publicly available dataset* to a target ML model through a predictive API with an aim *to create an informative and distributionally equivalent replica* of the target. First, we define *distributionally equivalent* and *Max-Information model extraction* attacks, and reduce them into a variational optimisation problem. The attacker sequentially solves this optimisation problem to select the most informative queries that simultaneously maximise the entropy and reduce the mismatch between the target and the stolen models. This leads to *an active sampling-based query selection algorithm*, MARICH, which is *model-oblivious*. Then, we evaluate MARICH on different text and image data sets, and different models, including CNNs and BERT. MARICH extracts models that achieve $\sim 60 - 95\%$ of true model's accuracy and uses $\sim 1,000 - 8,500$ queries from the publicly available datasets, which are different from the private training datasets. Models extracted by MARICH yield prediction distributions, which are $\sim 2 - 4\times$ closer to the target's distribution in comparison to the existing active sampling-based attacks. The extracted models also lead to 84-96% accuracy under membership inference attacks. Experimental results validate that MARICH is *query-efficient*, and capable of performing task-accurate, high-fidelity, and informative model extraction.

## 1 Introduction

In recent years, Machine Learning as a Service (MLaaS) is widely deployed and used in industries. In MLaaS [RGC15], an ML model is trained remotely on a private dataset, deployed in a Cloud, and offered for public access through a prediction API, such as Amazon AWS, Google API, Microsoft Azure. An API allows an user, including a potential *adversary*, *to send queries to the ML model and fetch corresponding predictions*. Recent works have shown such models with public APIs can be stolen, or extracted, by designing black-box model extraction attacks [TZJ+16]. In model extraction attacks, an adversary queries the target model with a query dataset, which might be same or different than the private dataset, collects the corresponding predictions from the target model, and builds a

---

[*]A significant portion of the work has been done as a part of P. Karmakar's masters in Ramakrishna Mission Vivekananda Educational and Research Institute, Belur, India.

[**] Code is available at: `https://github.com/debabrota-basu/marich`

37th Conference on Neural Information Processing Systems (NeurIPS 2023).

replica model of the target model. The goal is to construct a model which is almost-equivalent to the target model over input space [JCB+20].

Often, ML models are proprietary, guarded by IP rights, and expensive to build. These models might be trained on datasets which are expensive to obtain [YDY+19] and consist of private data of individuals [LM05]. Also, extracted models can be used to perform other privacy attacks on the private dataset used for training, such as membership inference [NSH19]. Thus, understanding susceptibility of models accessible through MLaaS presents an important conundrum. This motivates us to *investigate black-box model extraction attacks while the adversary has no access to the private data or a perturbed version of it* [PMG+17]. Instead, *the adversary uses a public dataset to query the target model* [OSF19, PGS+20].

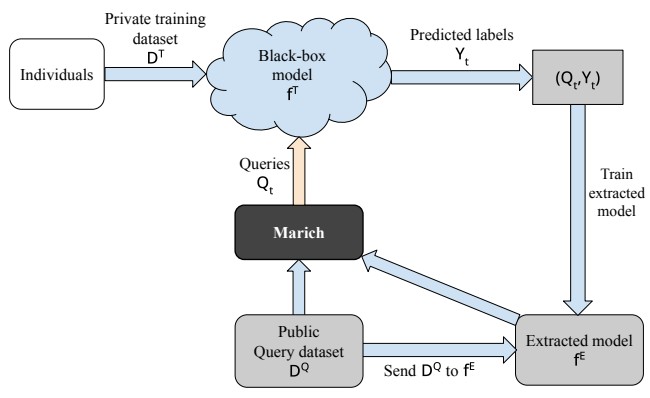

Figure 1: Black-box model extraction with MARICH.

Query-efficient black-box model extraction poses a tension between the number of queries sent to the target model and the accuracy of extracted model [PGS+20]. With more queries and predictions, an adversary can build a better replica. But querying an API too much can be expensive, as each query incurs a monetary cost in MLaaS. Also, researchers have developed algorithms that can detect adversarial queries, when they are not well-crafted or sent to the API in large numbers [JSMA19, PGKS21]. Thus, designing a query-efficient attack is paramount for practical deployment. Also, it exposes how more information can be leaked from a target model with less number of interactions.

In this paper, *we investigate effective definitions of efficiency of model extraction and corresponding algorithm design for query-efficient black-box model extraction attack with public data, which is oblivious to deployed model and applicable for any datatype.*

**Our contributions.** Our investigation yields three contributions.

*1. Formalism: Distributional equivalence and Max-Information extraction.* Often, the ML models, specifically classifiers, are stochastic algorithms. They also include different elements of randomness during training. Thus, rather than focusing on equivalence of extracted and target models in terms of a fixed dataset or accuracy on that dataset [JCB+20], we propose *a distributional notion of equivalence*. We propose that if the joint distribution induced by a query generating distribution and corresponding prediction distribution due to both the target and the extracted models are same, they will be called distributionally equivalent (Sec. 3). Another proposal is to reinforce the objective of the attack, i.e. to extract as much information as possible from the target model. This allows us to formulate the Max-Information attack, where the adversary aims to maximise the mutual information between the extracted and target models' distributions. *Our hypothesis is that if we can extract the predictive distribution of a model, it would be enough to replicate other properties of the model (e.g. accuracy) and also to run other attacks (e.g. membership inference), rather than designing specific attacks to replicate the task accuracy or the model weights.* We show that both the attacks can be performed by sequentially solving a single variational optimisation [SB12] problem (Eqn. (6)).

*2. Algorithm: Adaptive query selection for extraction with* MARICH. We propose an algorithm, MARICH (Sec. 4), that optimises the objective of the variational optimisation problem (Eqn. (6)). Given an extracted model, a target model, and previous queries, MARICH adaptively selects a batch of queries enforcing this objective. Then, it sends the queries to the target model, collects the predictions (i.e. the class predicted by target model), and uses them to further train the extracted model (Algo. 1). In order to select the most informative set of queries, it deploys three sampling strategies in cascade. These strategies select: a) the most informative set of queries, b) the most diverse set of queries in the first selection, and c) the final subset of queries where the target and extracted models mismatch the most. Together these strategies allow MARICH to select a small subset of queries that both maximise the information leakage, and align the extracted and target models (Fig. 1).

*3. Experimental analysis.* We perform extensive the most for a given modelevaluation with both image and text datasets, and diverse model classes, such as Logistic Regression (LR), ResNet, CNN, and BERT (Sec. 5). Leveraging MARICH's model-obliviousness, we even extract a ResNet trained on CIFAR10 with a CNN and out-of-class queries from ImageNet. Our experimental results validate that MARICH extracts more accurate replicas of the target model and high-fidelity replica of the target's prediction distributions in comparison to existing active sampling algorithms. While MARICH uses a small number of queries ($\sim 1k - 8.5k$) selected from publicly available query datasets, the extracted models yield accuracy comparable with the target model while encountering a membership inference attack. This shows that MARICH can extract alarmingly informative models query-efficiently. Additionally, as MARICH can extract the true model's predictive distribution with a different model architecture and a mismatched querying dataset, it allows us to design a model-oblivious and dataset-oblivious approach to attack.

## 1.1 Related works

**Taxonomy of model extraction.** Black-box model extraction (or model stealing or model inference) attacks aim to *replicate* of a target ML model, commonly classifiers, deployed in a remote service and accessible through a public API [TZJ+16]. The replication is done in such a way that the extracted model achieves one of the three goals: a) *accuracy close to that of the target model on the private training data* used to train the target model, b) *maximal agreement in predictions* with the target model on the *private training data*, and c) maximal agreement in prediction with the target model over the *whole input domain*. Depending on the objective, they are called *task accuracy*, *fidelity*, and *functional equivalence model extractions*, respectively [JCB+20]. Here, *we generalise these three approaches using a novel definition of distributional equivalence and also introduce a novel information-theoretic objective of model extraction which maximises the mutual information between the target and the extracted model over the whole data domain.*

**Frameworks of attack design.** Following [TZJ+16], researchers have proposed multiple attacks to perform one of the three types of model extraction. The attacks are based on two main approaches: *direct recovery* (target model specific) [MSDH19, BBJP18, JCB+20] and *learning* (target model specific/oblivious). The learning-based approaches can also be categorised into supervised learning strategies, where the adversary has access to both the true labels of queries and the labels predicted by the target model [TZJ+16, JCB+20], and online active learning strategies, where the adversary has only access to the predicted labels of the target model, and actively select the future queries depending on the previous queries and predicted labels [PMG+17, PGS+20, CCG+20]. *As query-efficiency is paramount for an adversary while attacking an API to save the budget and to keep the attack hidden and also the assumption of access true label from the private data is restrictive, we focus on designing an online and active learning-based attack strategy that is model oblivious.*

**Types of target model.** While [MSDH19, CCG+20] focus on performing attacks against linear models, all others are specific to neural networks [MSDH19, JCB+20, PGS+20] and even a specific architecture [CSBB+18]. In contrast, MARICH is *capable of attacking both linear models and neural networks*. Additionally, MARICH *is model-oblivious*, i.e. it can attack one model architecture (e.g. ResNet) using a different model architecture (e.g. CNN).

**Types of query feedback.** Learning-based attacks often assume access to either the probability vector of the target model over all the predicted labels [TZJ+16, OSF19, PGS+20, JCB+20], or the gradient of the last layer of the target neural network [MSDH19, MHS21], which are hardly available in a public API. In contrast, following [PMG+17], *we assume access to only the predicted labels of the target model for a set of queries*, which is always available with a public API. Thus, experimentally, we cannot compare with existing active sampling attacks requiring access to the whole prediction vector [PGS+20, OSF19], and thus, compare with a wide-range of active sampling methods that can operate only with the predicted label, such as $K$-center sampling, entropy sampling, least confidence sampling, margin sampling etc. [RXC+21]. Details are in Appendix C.

**Choices of public datasets for queries.** There are two approaches of querying a target model: *data-free* and *data-selection based*. In *data-free attacks*, the attacker begins with noise. The informative queries are generated further using a GAN-like model fed with responses obtained from an API [ZWL+20, TMWP21, MHS21, ZLX+22, SAB22]. Typically, it requires almost a million queries to the API to start generating sensible query data (e.g. sensible images that can leak from a model trained on CIFAR10). But since one of our main focus is query-efficiency, we focus on *data-selection based attacks*, where an adversary has access to a query dataset to select the queries

from and to send it to the target model to obtain predicted labels. In literature, researchers assume three types of query datasets: *synthetically generated samples* [TZJ+16], *adversarially perturbed private (or task domain) dataset* [PMG+17, JSMA19], and *publicly available (or out-of-task domain) dataset* [OSF19, PGS+20]. As we do not want to restrict MARICH to have access to the knowledge of the private dataset or any perturbed version of it, *we use publicly available datasets, which are different than the private dataset.* To be specific, we only assume whether we should query the API with images, text, or tabular data and not even the identical set of labels. For example, we experimentally attack models trained on CIFAR10 with ImageNet queries having different classes.

Further discussions on related active sampling algorithms and distinction of MARICH with the existing works are deferred to Appendix C.

## 2 Background: Classifiers, model extraction, membership inference attacks

Before proceeding to the details, we present the fundamentals of a classifier in ML, and two types of inference attacks: Model Extraction (ME) and Membership Inference (MI).

**Classifiers.** A classifier in ML [GBCB16] is a function $f : \mathcal{X} \to \mathcal{Y}$ that maps a set of input features $\mathbf{X} \in \mathcal{X}$ to an output $Y \in \mathcal{Y}$.[2] The output space is a finite set of classes, i.e. $\{1, \ldots, k\}$. Specifically, a classifier $f$ is a parametric function, denoted as $f_\theta$, with parameters $\theta \in \mathbb{R}^d$, and is trained on a dataset $\mathbf{D}^T$, i.e. a collection of $n$ tuples $\{(\mathbf{x}_i, y_i)\}_{i=1}^n$ generated IID from an underlying distribution $\mathcal{D}$. Training implies that given a model class $\mathcal{F} = \{f_\theta | \theta \in \Theta\}$, a loss function $l : \mathcal{Y} \times \mathcal{Y} \to \mathbb{R}_{\geq 0}$, and training dataset $\mathbf{D}^T$, we aim to find the optimal parameter $\theta^* \triangleq \arg \min_{\theta \in \Theta} \sum_{i=1}^n l(f_\theta(\mathbf{x}_i), y_i)$. We use cross-entropy, i.e. $l(f_\theta(\mathbf{x}_i), y_i) \triangleq -y_i \log(f_\theta(\mathbf{x}_i))$, as the loss function for classification.

**Model extraction attack.** A model extraction attack is an inference attack where an adversary aims to steal a target model $f^T$ trained on a private dataset $\mathbf{D}^T$ and create another replica of it $f^E$ [TZJ+16]. In the black-box setting that we are interested in, the adversary can only query the target model $f^T$ by sending queries $Q$ through a publicly available API and to use the corresponding predictions $\hat{Y}$ to construct $f^E$. The goal of the adversary is to create a model which is either (a) as similar to the target model as possible for all input features, i.e. $f^T(x) = f^E(x) \, \forall x \in \mathcal{X}$ [SS20, CCG+20] or (b) predicts labels that has maximal agreement with that of the labels predicted by the target model for a given data-generating distribution, i.e. $f^E = \arg \min \Pr_{x \sim \mathcal{D}}[l(f^E(x), f^T(x))]$ [TZJ+16, PGS+20, JCB+20]. The first type of attacks are called the functionally equivalent attacks. The later family of attacks is referred as the fidelity extraction attacks. The third type of attacks aim to find an extracted model $f^E$ that achieves maximal classification accuracy for the underlying private dataset used to train the $f^T$. These are called task accuracy extraction attacks [TZJ+16, MSDH19, OSF19]. In this paper, *we generalise the first two type of attacks by proposing the distributionally equivalent attacks and experimentally show that it yields both task accuracy and fidelity.*

**Membership inference attack.** Another popular family of inference attacks on ML models is the Membership Inference (MI) attacks [SSSS17, YGFJ18]. In MI attack, given a private (or member) dataset $\mathbf{D}^T$ to train $f^T$ and another non-member dataset $S$ with $|\mathbf{D}^T \cap S| \neq \emptyset$, the goal of the adversary is to infer whether any $x \in \mathcal{X}$ is sampled from the member dataset $\mathbf{D}^T$ or the non-member dataset $S$. Effectiveness of an MI attacks can be measured by its accuracy of MI, i.e. the total fraction of times the MI adversary identifies the member and non-member data points correctly. Accuracy of MI attack on the private data using $f^E$ rather than $f^T$ is considered as a measure of effectiveness of the extraction attack [NSH19]. We show that the model $f^E$ extracted using MARICH allows us to obtain similar MI accuracy as that obtained by directly attacking the target model $f^T$ using even larger number of queries. This validates that *the model $f^E$ by MARICH in a black-box setting acts as an information equivalent replica of the target model $f^T$.*

## 3 Distributional equivalence and Max-Information model extractions

In this section, we introduce the distributionally equivalent and Max-Information model extractions. We further reduce both the attacks into a variational optimisation problem.

**Definition 3.1 (Distributionally equivalent model extraction).** For any query generating distribution $\mathcal{D}^Q$ over $\mathbb{R}^d \times \mathcal{Y}$, an extracted model $f^E : \mathbb{R}^d \to Y$ is distributionally equivalent to a target

---

[2]We denote sets/vectors by **bold** letters, and the distributions by *calligraphic* letters. We express random variables in UPPERCASE, and an assignment of a random variable in lowercase.

model $f^T : \mathbb{R}^d \to Y$, if the joint distributions of input features $Q \in \mathbb{R}^d \sim \mathcal{D}^Q$ and predicted labels induced by both the models are same almost surely. This means that for any divergence $D$, two distributionally equivalent models $f^E$ and $f^T$ satisfy $D(\Pr(f^T(Q), Q) \| \Pr(f^E(Q), Q)) = 0 \, \forall \, \mathcal{D}^Q$.

To ensure query-efficiency in distributionally equivalent model extraction, an adversary aims to choose a query generating distribution $\mathcal{D}^Q$ that minimises it further. If we assume that the extracted model is also a parametric function, i.e. $f_\omega^E$ with parameters $\omega \in \Omega$, we can solve the query-efficient distributionally equivalent extraction by computing

$$(\omega_{\text{DEq}}^*, \mathcal{D}_{\min}^Q) \triangleq \underset{\omega \in \Omega}{\arg\min} \, \underset{\mathcal{D}^Q}{\arg\min} \, D(\Pr(f_{\theta^*}^T(Q), Q) \| \Pr(f_\omega^E(Q), Q)). \tag{1}$$

Equation (1) allows us to choose a different class of models with different parametrisation for extraction till the joint distribution induced by it matches with that of the target model. For example, the extracted model can be a logistic regression or a CNN if the target model is a logistic regression. This formulation also enjoys the freedom to choose the data distribution $\mathcal{D}^Q$ for which we want to test the closeness. Rather the distributional equivalence pushes us to find the best query distribution for which the mismatch between the posteriors reduces the most and to compute an extracted model $f_{\omega^*}^E$ that induces the joint distribution closest to that of the target model $f_{\theta^*}^T$.

**Connection with different types of model extraction.** For $D = D_{\text{KL}}$, our formulation extends the fidelity extraction from label agreement to prediction distribution matching, which addresses the future work indicated by [JCB+20]. If we choose $\mathcal{D}_{\min}^Q = \mathcal{D}^T$, and substitute $D$ by prediction agreement, distributional equivalence retrieves the fidelity extraction attack. If we choose $\mathcal{D}_{\min}^Q = \text{Unif}(\mathcal{X})$, distributional equivalent extraction coincides with functional equivalent extraction. Thus, a distributional equivalence attack can lead to both fidelity and functional equivalence extractions depending on the choice of query generating distribution $\mathcal{D}^Q$ and the divergence $D$.

**Theorem 3.2** (Upper bounding distributional closeness). *If we choose KL-divergence as the divergence function $D$, then for a given query generating distribution $\mathcal{D}^Q$*

$$D_{\text{KL}}(\Pr(f_{\theta^*}^T(Q), Q) \| \Pr(f_{\omega_{\text{DEq}}^*}^E(Q), Q)) \leq \min_\omega \mathbb{E}_Q[l(f_{\theta^*}^T(Q), f_\omega^E(Q))] - H(f_\omega^E(Q)). \tag{2}$$

By variational principle, Theorem 3.2 implies that *minimising the upper bound* on the RHS leads to an extracted model which minimises the KL-divergence for a chosen query distribution.

**Max-Information model extraction.** Objective of any inference attack is to leak as much information as possible from the target model $f^T$. Specifically, in model extraction attacks, we want to create an informative replica $f^E$ of the target model $f^T$ such that it induces a joint distribution $\Pr(f_\omega^E(Q), Q)$, which retains the most information regarding the target's joint distribution. As adversary controls the query distribution, we aim to choose a query distribution $\mathcal{D}^Q$ that maximises information leakage.

**Definition 3.3** (**Max-Information model extraction**). A model $f^E : \mathbb{R}^d \to Y$ and a query distribution $\mathcal{D}^Q$ are called a Max-Information extraction of a target model $f^T : \mathbb{R}^d \to Y$ and a Max-Information query distribution, respectively, if they maximise the mutual information between the joint distributions of input features $Q \in \mathbb{R}^d \sim \mathcal{D}^Q$ and predicted labels induced by $f^E$ and that of the target model. Mathematically, $(f_{\omega^*}^E, \mathcal{D}_{\max}^Q)$ is a Max-Information extraction of $f_{\theta^*}^T$ if

$$(\omega_{\text{MaxInf}}^*, \mathcal{D}_{\max}^Q) \triangleq \underset{\omega}{\arg\max} \, \underset{\mathcal{D}_Q}{\arg\max} \, I(\Pr(f_{\theta^*}^T(Q), Q) \| \Pr(f_\omega^E(Q), Q)) \tag{3}$$

Similar to Definition 3.1, Definition 3.3 also does not restrict us to choose a parametric model $\omega$ different from that of the target $\theta$. It also allows us to compute the data distribution $\mathcal{D}^Q$ for which the information leakage is maximum rather than relying on the private dataset $\mathbf{D}^T$ used for training $f^T$.

**Theorem 3.4** (Lower bounding information leakage). *For any given distribution $\mathcal{D}^Q$, the information leaked by any Max-Information attack (Equation 3) is lower bounded as:*

$$I(\Pr(f_{\theta^*}^T(Q), Q) \| \Pr(f_{\omega_{\text{MaxInf}}^*}^E(Q), Q)) \geq \max_\omega -\mathbb{E}_Q[l(f_{\theta^*}^T(Q), f_\omega^E(Q))] + H(f_\omega^E(Q)). \tag{4}$$

By variational principle, Theorem 3.4 implies that *maximising the lower bound* in the RHS will lead to an extracted model which maximises the mutual information between target and extracted joint distributions for a given query generating distribution.

**Distributionally equivalent and Max-Information extractions: A variational optimisation formulation.** From Theorem 3.2 and 3.4, we observe that the lower and upper bounds of the objective functions of distribution equivalent and Max-Information attacks are negatives of each other. Specifically, $-D_{\mathrm{KL}}(\Pr(f_{\theta^*}^T(Q), Q) \| \Pr(f_{\omega_{\mathrm{DEq}}^*}^E(Q), Q)) \geq \max_\omega -F(\omega, \mathcal{D}^Q)$ and $I(\Pr(f_{\theta^*}^T(Q), Q) \| \Pr(f_{\omega_{\mathrm{MaxInf}}^*}^E(Q), Q)) \geq \max_\omega F(\omega, \mathcal{D}^Q)$, where

$$F(\omega, \mathcal{D}^Q) \triangleq -\mathbb{E}_Q[l(f_{\theta^*}^T(Q), f_\omega^E(Q))] + H(f_\omega^E(Q)). \tag{5}$$

Thus, following a variational approach, we aim to solve an optimisation problem on $F(\omega, \mathcal{D}^Q)$ in an online and frequentist manner. We do not assume a parametric family of $\mathcal{D}^Q$. Instead, we choose a set of queries $Q_t \in \mathbb{R}^d$ at each round $t \in T$. This leads to an empirical counterpart of our problem:

$$\max_{\omega \in \omega, Q_{[0,T]} \in \mathbf{D}^Q_{[T]}} \hat{F}(\omega, Q_{[0,T]}) \triangleq \max_{\omega, Q_{[0,T]}} -\frac{1}{T} \sum_{t=1}^T l(f_{\theta^*}^T(Q_t), f_\omega^E(Q_t))] + \sum_{t=1}^T H(f_\omega^E(Q_t)). \tag{6}$$

As we need to evaluate $f_{\theta^*}^T$ for each $Q_t$, we refer $Q_t$'s as *queries*, the dataset $\mathbf{D}^Q \subseteq \mathbb{R}^d \times \mathcal{Y}$ from where they are chosen as the *query dataset*, and the corresponding unobserved distribution $\mathcal{D}^Q$ as the *query generating distribution*. Given the optimisation problem of Equation 6, we propose an algorithm MARICH to solve it effectively.

## 4 Marich: A query selection algorithm for model extraction

In this section, we propose an algorithm, MARICH, to solve Equation (6) in an adaptive manner.

**Algorithm design.** We observe that once the queries $Q_{[0,T]}$ are selected, the outer maximisation problem of Eq. (6) is equivalent to regualrised loss minimisation. Thus, it can be solved using any standard empirical risk minimisation algorithm (e.g. Adam, SGD). Thus, to achieve query efficiency, we focus on designing a query selection algorithm that selects a batch of queries $Q_t$ at round $t \leq T$:

$$Q_t \triangleq \arg\max_{Q \in \mathbf{D}^Q} \underbrace{-\frac{1}{t} \sum_{i=1}^{t-1} l(f_{\theta^*}^T(Q_i \cup Q), f_{\omega_{t-1}}^E(Q_i \cup Q))]}_{\text{Model-mismatch term}} + \underbrace{\sum_{i=1}^{t-1} H(f_{\omega_{t-1}}^E(Q_i \cup Q))}_{\text{Entropy term}}. \tag{7}$$

---

**Algorithm 1** MARICH

---

**Input**: Target model: $f^T$, Query dataset: $\mathbf{D}^Q$, #Classes: $k$
**Parameter**: #initial samples: $n_0$, Training epochs: $E_{max}$, #Batches of queries: $T$, Query budget: $B$, Subsampling ratios: $\gamma_1, \gamma_2 \in (0, 1]$
**Output**: Extracted model $f^E$

1: //* Initialisation of the extracted model*//                                      ▷ *Phase 1*
2: $Q_0^{train} \leftarrow n_0$ datapoints randomly chosen from $D^Q$
3: $Y_0^{train} \leftarrow f^T(Q_0^{train})$                           ▷ *Query the target model $f^T$ with $Q_0^{train}$*
4: **for** epoch $\leftarrow 1$ to $E_{max}$ **do**
5:     $f_0^E \leftarrow$ Train $f^E$ with $(Q_0^{train}, Y_0^{train})$
6: **end for**
7: //* Adaptive query selection to build the extracted model*//               ▷ *Phase 2*
8: **for** $t \leftarrow 1$ to $T$ **do**
9:     $Q_t^{entropy} \leftarrow$ ENTROPYSAMPLING$(f_{t-1}^E, \mathbf{D}^Q \setminus Q_{t-1}^{train}, B)$
10:     $Q_t^{grad} \leftarrow$ ENTROPYGRADIENTSAMPLING$(f_{t-1}^E, Q_t^{entropy}, \gamma_1 B)$
11:     $Q_t^{loss} \leftarrow$ LOSSSAMPLING$(f_{t-1}^E, Q_t^{grad}, Q_{t-1}^{train}, Y_{t-1}^{train}, \gamma_1\gamma_2 B)$
12:     $Y_t^{new} \leftarrow f^T(Q_t^{loss})$                       ▷ *Query the target model $f^T$ with $Q_t^{loss}$*
13:     $Q_t^{train} \leftarrow Q_{t-1}^{train} \cup Q_t^{loss}$
14:     $Y_t^{train} \leftarrow Y_{t-1}^{train} \cup Y_t^{new}$
15:     **for** epoch $\leftarrow 1$ to $E_{max}$ **do**
16:         $f_t^E \leftarrow$ Train $f_{t-1}^E$ with $(Q_t^{train}, Y_t^{train})$
17:     **end for**
18: **end for**
19: **return** Extracted model $f^E \leftarrow f_T^E$

---

Here, $f^E_{\omega_{t-1}}$ is the model extracted by round $t-1$. Equation (7) indicates two criteria to select the queries. With the **entropy term**, we want to select a query that maximises the entropy of predictions for the extracted model $f^E_{\omega_{t-1}}$. This allows us to select the queries which are most informative about the mapping between the input features and the prediction space. With the **model-mismatch term**, Eq. (7) pushes the adversary to select queries where the target and extracted models mismatch the most. Thus, minimising the loss between target and extracted models for such a query forces them to match over the whole domain. Algorithm 1 illustrates a pseudocode of MARICH (Appendix A).

**Initialisation phase.** To initialise the extraction, we select a set of $n_0$ queries, called $Q^{train}_0$, uniformly randomly from the query dataset $\mathbf{D}^Q$. We send these queries to the target model and collect corresponding predicted classes $Y^{train}_0$ (Line 3). We use these $n_0$ samples of input-predicted label pairs to construct a primary extracted model $f^E_0$.

**Active sampling.** As the adaptive sampling phase commences, we select $\gamma_1\gamma_2 B$ number of queries at round $t$. To *maximise* the **entropy term** and *minimise* the **model-mismatch term** of Eq. (7), we sequentially deploy ENTROPYSAMPLING and LOSSSAMPLING. To achieve further query-efficiency, we refine the queries selected using ENTROPYSAMPLING by ENTROPYGRADIENTSAMPLING, which finds the most diverse subset from a given set of queries. Now, we describe the sampling strategies.

ENTROPYSAMPLING. First, we aim to select the set of queries which unveil most information about the mapping between the input features and the prediction space. Thus, we deploy ENTROPYSAMPLING. In ENTROPYSAMPLING, we compute the output probability vectors from $f^E_{t-1}$ for all the query points in $\mathbf{D}^Q \setminus Q^{train}_{t-1}$ and then select top $B$ points with highest entropy:

$$Q^{entropy} \leftarrow \underset{X \subset X_{in}, |X|=B}{\arg\max} H(f^E(X_{in})).$$

Thus, we select the queries $Q^{entropy}_t$, about which $f^E_{t-1}$ is most confused and training on these points makes the model more informative.

ENTROPYGRADIENTSAMPLING. To be frugal about the number of queries, we refine $Q^{entropy}_t$ to compute the most diverse subset of it. First, we compute the gradients of entropy of $f^E_{t-1}(x)$, i.e. $\nabla_x H(f^E_{t-1}(x))$, for all $x \in Q^{entropy}_t$. The gradient at point $x$ reflects the change at $x$ in the prediction distribution induced by $f^E_{t-1}$. We use these gradients to embed the points $x \in Q^{entropy}_t$. Now, we deploy K-means clustering to find $k$ (= #classes) clusters with centers $C_{in}$. Then, we sample $\gamma_1 B$ points from these clusters:

$$Q^{grad} \leftarrow \underset{X \subset Q^{entropy}_t, |X|=\gamma_1 B}{\arg\min} \sum_{x_i \in X} \sum_{x_j \in C_{in}} \|\nabla_{x_i} H(f^E(.)) - \nabla_{x_j} H(f^E(.))\|^2_2.$$

Selecting from $k$ clusters ensures diversity of queries and reduces them by $\gamma_1$.

LOSSSAMPLING. We select points from $Q^{grad}_t$ for which the predictions of $f^T_{\theta^*}$ and $f^E_{t-1}$ are most dissimilar. To identify these points, we compute the loss $l(f^T(x), f^E_{t-1}(x))$ for all $x \in Q^{train}_{t-1}$. Then, we select top-$k$ points from $Q^{train}_{t-1}$ with the highest loss values (Line 11), and sample a subset $Q^{loss}_t$ of size $\gamma_1\gamma_2 B$ from $Q^{grad}_t$ which are closest to the $k$ points selected from $Q^{train}_{t-1}$. This ensures that $f^E_{t-1}$ would better align with $f^T$ if it trains on the points where the mismatch in predictions are higher.

At the end of Phase 2 in each round of sampling, $Q^{loss}_t$ is sent to $f^T$ for fetching the labels $Y^{train}_t$ predicted by the target model. We use $(Q^{loss}_t, Y^{loss}_t)$ along with $(Q^{train}_{t-1}, Y^{train}_{t-1})$ to train $f^E_{t-1}$ further. Thus, MARICH performs $n_0 + \gamma_1\gamma_2 BT$ number of queries through $T+1$ number of interactions with the target model $f^T$ to create the final extracted model $f^E_T$. We experimentally demonstrate effectiveness of the model extracted by MARICH to achieve high task accuracy and to act as an informative replica of the target for extracting private information regarding private training data $\mathbf{D}^T$.

**Discussions.** Eq. (7) dictates that the active sampling strategy should try to select queries that maximise the entropy in the prediction distribution of the extracted model, while decreases the mismatch in predictions of the target and the extracted models. We further use the ENTROPYGRADIENTSAMPLING to choose a smaller but most diverse subset. As Eq. (7) does not specify any ordering between these objectives, one can argue about the sequence of using these three sampling strategies. We choose to use sampling strategies in the decreasing order of runtime complexity as the first strategy selects the queries from the whole query dataset, while the following strategies work only on the

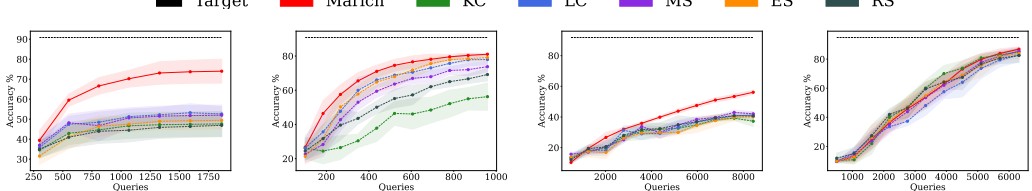

| (a) LR/EMNIST query | (b) LR/CIFAR10 query | (c) ResNet/ImgNet query | (d) CNN/EMNIST query |

Figure 2: Accuracy of the extracted models (mean ± std. over 10 runs) w.r.t. the target model using MARICH, and competing active sampling methods (KC, LC, MS, ES, RS). Each figure represents (a target model, a query dataset). Models extracted by MARICH are closer to the target models.

already selected queries. We show in Appendix E that LOSSSAMPLING incurs the highest runtime followed by ENTROPYGRADIENTSAMPLING, while ENTROPYSAMPLING is significantly cheaper.

*Remark* 4.1. Previously, researchers have deployed different active sampling methods to efficiently select queries for attacks [PMG+17, CCG+20, PGS+20]. But our derivation shows that the active query selection can be grounded on the objectives of distributionally equivalent and max-information extractions. Thus, though the end result of our formulation, i.e. MARICH, is an active query selection algorithm, our framework is different and novel with respect to existing active sampling based works.

## 5 Experimental analysis

Now, we perform an experimental evaluation of models extracted by MARICH. Here, we discuss the experimental setup, the objectives of experiments, and experimental results. We defer the source code, extended results, parametric similarity of the extracted models, effects of model-mismatch, details of different samplings, and hyperparameters to Appendix.

**Experimental setup.** We implement a prototype of MARICH using Python 3.9 and PyTorch 1.12, and run on a NVIDIA GeForce RTX 3090 24 GB GPU. We perform attacks against *four target models* ($f^T$), namely Logistic Regression (LR), CNN [LBH15], ResNet [HZRS16], BERT [DCLT18], trained on *three private datasets* ($\mathbf{D}^T$): MNIST handwritten digits [Den12], CIFAR10 [KH+09] and BBC News, respectively. For model extraction, we use EMNIST letters dataset [CATvS17], CIFAR10, ImageNet [DDS+09], and AGNews [ZZL15], as publicly-available, mismatched query datasets $\mathbf{D}^Q$.

To instantiate task accuracy, we compare accuracy of the extracted models $f^E_{\text{MARICH}}$ with the target model and models extracted by K-Center (KC) [SS18], Least-Confidence sampling (LC) [LS06], Margin sampling (MS) [BBZ07, JG19], Entropy Sampling (ES) [LG94], and Random Sampling (RS). To instantiate informativeness of the extracted models [NSH19], we compare the Membership Inference (MI), i.e. MI accuracy and MI agreements (% and AUC), performed on the target models, and the models extracted using MARICH and competitors with same query budget. For MI, we use in-built membership attack from IBM ART [NST+18]. For brevity, we discuss Best of Competitors (BoC) against MARICH for each experiment (except Fig. 2- 3) The objectives of the experiments are:

1. *How do the accuracy of the model extracted using* MARICH *on the private dataset compare with that of the target model, and RS with same query budget?*

2. *How close are the prediction distributions of the model extracted using* MARICH *and the target model? Can* MARICH *produce better replica of target's prediction distribution than other active sampling methods, leading to better distributional equivalence?*

3. *How do the models extracted by* MARICH *behave under Membership Inference (*MI*) in comparison to the target models, and the models extracted by RS with same budget?* The MI accuracy achievable by attacking a model acts as a proxy of how informative is the model.

4. *How does the performance of extracted models change if Differentially Private (DP) mechanisms [DMNS06] are applied on target model either during training or while answering the queries?*

**Accuracy of extracted models.** MARICH extracts LR models with 1,863 and 959 queries selected from EMNIST and CIFAR10, while attacking a target LR model, $f^T_{\text{LR}}$ trained on MNIST (test accuracy: 90.82%). The models extracted by MARICH using EMNIST and CIFAR10 achieve test accuracy 73.98% and 86.83% (81.46% and 95.60% of $f^T_{\text{LR}}$), respectively (Figure 2a-2b). The models extracted using BoC show test accuracy 52.60% and 79.09% (57.91% and 87.08% of $f^T_{\text{LR}}$), i.e.

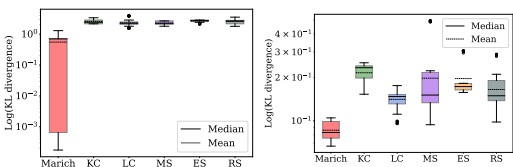
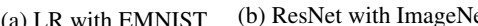
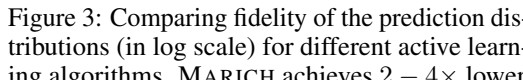

(a) LR with EMNIST    (b) ResNet with ImageNet

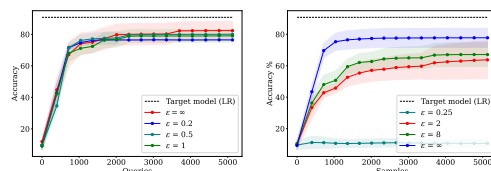

(a) DP-SGD to train target (b) Perturb output of query

Figure 3: Comparing fidelity of the prediction distributions (in log scale) for different active learning algorithms. MARICH achieves $2 - 4\times$ lower KL-divergence than others.

Figure 4: Comparing test accuracy of the models extracted by MARICH against different DP mechanisms (DP-SGD and Output Perturbation) applied on the target model.

Table 1: Statistics of accuracy & membership inference (MI) for different target models, datasets & attacks. "-" means member dataset and target model is used. *BoC means Best of Competitors.

| Member dataset | Target model | Query Dataset | Algorithm | Non-member dataset | #Queries | MI acc. | MI agreement | MI agreement AUC | Accuracy |
|---|---|---|---|---|---|---|---|---|---|
| MNIST | LR | - | - | EMNIST | 50,000 (100%) | 87.99% | - | - | 90.82% |
| MNIST | LR | EMNIST | MARICH | EMNIST | 1863 (3.73%) | 84.47% | 90.34% | 90.89% | 73.98% |
| MNIST | LR | EMNIST | BoC* | EMNIST | 1863 (3.73%) | 78.00% | 80.11% | 83.07% | 52.60% |
| MNIST | LR | - | - | CIFAR10 | 50,000 (100%) | 98.02% | - | - | 90.82% |
| MNIST | LR | CIFAR10 | MARICH | CIFAR10 | 959 (1.92%) | 96.32% | 96.89% | 94.32% | 81.06% |
| MNIST | LR | CIFAR10 | BoC* | CIFAR10 | 959 (1.92%) | 93.70% | 93.67% | 91.53% | 77.93% |
| MNIST | CNN | - | - | EMNIST | 50,000 (100%) | 89.97% | - | - | 94.83% |
| MNIST | CNN | EMNIST | MARICH | EMNIST | 6317 (12.63%) | 90.62% | 87.27% | 86.71% | 86.83% |
| MNIST | CNN | EMNIST | BoC* | EMNIST | 6317 (12.63%) | 90.73% | 87.53% | 86.97% | 82.51% |
| CIFAR10 | ResNet | - | - | EMNIST | 50,000 (100%) | 93.61% | - | - | 91.82% |
| CIFAR10 | ResNet | ImageNet | MARICH | EMNIST | 8429 (16.58%) | 90.40% | 93.84% | 76.51% | 56.11% |
| CIFAR10 | ResNet | ImageNet | BoC* | EMNIST | 8429 (16.58%) | 90.08% | 95.41% | 72.94% | 40.66% |
| BBCNews | BERT | - | - | AGNews | 1,490 (100%) | 98.61% | - | - | 98.65% |
| BBCNews | BERT | AGNews | MARICH | AGNews | 1,070 (0.83%) | 94.42% | 91.02% | 82.62% | 87.01% |
| BBCNews | BERT | AGNews | BoC* | AGNews | 1,070 (0.83%) | 89.17% | 86.93% | 58.64% | 76.41% |

significantly less than that of MARICH. MARICH attacks a ResNet, $f^T_{\text{ResNet}}$, trained on CIFAR10 (test accuracy: $91.82\%$) with 8,429 queries from ImageNet dataset, and extracts a CNN. The extracted CNN shows $56.11\%$ ($61.10\%$ of $f^T_{ResNet}$) test accuracy. But the model extracted using BoC achieves $42.05\%$ ($45.79\%$ of $f^T_{ResNet}$) accuracy (Figure 2c). We also attack a CNN with another CNN, which also reflects MARICH's improved performance (Figure 2d). *To verify* MARICH*'s effectiveness for text data*, we also attack a BERT, $f^T_{BERT}$ trained on BBCNews (test accuracy: $98.65\%$) with queries from the AGNews dataset. By using only 474 queries, MARICH extracts a model with $85.45\%$ ($86.64\%$ of $f^T_{BERT}$) test accuracy. The model extracted by BoC shows test accuracy $79.25\%$ ($80.36\%$ of $f^T_{BERT}$). *For all the models and datasets,* MARICH *extracts models that achieve test accuracy closer to target models, and are more accurate than models extracted by the other algorithms.*

**Distributional equivalence of extracted models.** One of our aims is to extract a distributionally equivalent model of the target $f^T$ using MARICH. Thus, in Figure 3, we illustrate the KL-divergence (mean±std. over 10 runs) between the prediction distributions of the target model and the model extracted by MARICH. Due to brevity, we show two cases in the main paper: when we attack i) an LR trained on MNIST with EMNIST with an LR, and ii) a ResNet trained on CIFAR10 with ImageNet with a CNN. In all cases, we observe that the models extracted by MARICH achieve $\sim 2 - 4\times$ lower KL-divergence than the models extracted by all other active sampling methods. *These results show that* MARICH *is extracts high-fidelity distributionally equivalent models than competing algorithms.*

**Membership inference with extracted models.** In Table 1, we report *accuracy*, *agreement* in inference with target model, and *agreement AUC* of membership attacks performed on different target models and extracted models with different query datasets. The models extracted using MARICH demonstrate higher MI agreement with the target models than the models extracted using its competitors in most of the cases. They also achieve MI accuracy close to the target model. *These results indicate that the models extracted by* MARICH *act as informative replicas of the target models.*

**Performance against privacy defenses.** We test the impact of DP-based defenses deployed in the target model on the performance of MARICH. First, we train four target models on MNIST using *DP-SGD* [ACG+16] with privacy budgets $\varepsilon = \{0.2, 0.5, 1, \infty\}$ and $\delta = 10^{-5}$. As illustrated in Figure 4a, accuracy of the models extracted by querying DP target models are $\sim 2.3 - 7.4\%$ lower than the model extracted from non-private target models. Second, we apply an *output perturbation* method [DMNS06], where a calibrated Laplace noise is added to the responses of the target model against MARICH's queries. This ensures $\varepsilon$-DP for the target model. Figure 4b shows that performance of the extracted models degrade slightly for $\varepsilon = 2, 8$, but significantly for $\varepsilon = 0.25$. Thus, *performance of* MARICH *decreases while operating against DP defenses but the degradation varies depending on the defense mechanism.*

**Summary of results.** From the experimental results, we deduce the following conclusions.

*1. Accuracy.* Test accuracy (on the subsets of private datasets) of the models $f_{\text{MARICH}}^E$ are higher than the models extracted with the competing algorithms, and are $\sim 60 - 95\%$ of the target models (Fig. 2). This shows effectiveness of MARICH as a task accuracy extraction attack, while solving distributional equivalence and max-info extractions.

*2. Distributional equivalence.* We observe that the KL-divergence between the prediction distributions of the target model and $f_{\text{MARICH}}^E$ are $\sim 2-4\times$ lower than the models extracted by other active sampling algorithms. This confirms that MARICH conducts more accurate distributionally equivalent extraction than existing active sampling attacks.

*3. Informative replicas: Effective membership inference.* The agreement in MI achieved by attacking $f_{\text{MARICH}}^E$ and the target model in most of the cases is higher than that of the BoC* (Table 1). Also, MI accuracy for $f_{\text{MARICH}}^E$'s are $84.74\% - 96.32\%$ (Table 1). This shows that the models extracted by MARICH act as informative replicas of the target model.

*4. Query-efficiency.* Table 1 shows that MARICH uses only $959 - 8,429$ queries from the public datasets, i.e. a small fraction of data used to train the target models. Thus, MARICH is significantly query efficient, as existing active learning attacks use 10k queries to commence [PGS+20, Table 2].

*5. Performance against defenses.* Performance of MARICH decreases with the increasing level of DP applied on the target model, which is expected. But when DP-SGD is applied to train the target, the degradation is little ($\sim 7\%$) even for $\varepsilon = 0.2$. In contrast, the degradation is higher when the output perturbation is applied with similar $\varepsilon$ (0.25).

*6. Model-obliviousness and out-of-class data.* By construction, MARICH is model-oblivious and can use out-of-class public data to extract a target model. To test this flexibility of MARICH, we try and extract a ResNet trained on CIFAR10 using a different model, i.e. CNN, and out-of-class data, i.e. ImageNet. We show CNNs extracted by MARICH are more accurate, distributionally close, and also lead to higher MI accuracy that the competitors, validating flexibility of MARICH.

*7. Resilience to mismatch between $\mathbf{D}^T$ and $\mathbf{D}^Q$.* For MARICH, the datasets $\mathbf{D}^T$ and $\mathbf{D}^Q$ can be significantly different. For example, we attack an MNIST-trained model with EMNIST and CIFAR10 as query datasets. MNIST contains handwritten digits, CIFAR10 contains images of aeroplanes, cats, dogs etc., and EMNIST contains handwritten letters. Thus, the data generating distributions and labels are significantly different between the private and query datasets. We also attack a CIFAR10-trained ResNet with ImageNet as $\mathbf{D}^Q$. CIFAR10 and ImageNet are also known to have very different labels and images. Our experiments demonstrate that Marich can handle data mismatch as well as model mismatch, which is an addendum to the existing model extraction attacks.

## 6  Conclusion and future directions

We investigate the design of a model extraction attack against a target ML model (classifier) trained on a private dataset and accessible through a public API. The API returns only a predicted label for a given query. We propose the notions of distributional equivalence extraction, which extends the existing notions of task accuracy and functionally equivalent model extractions. We also propose an information-theoretic notion, i.e. Max-Info model extraction. We further propose a variational relaxation of these two types of extraction attacks, and solve it using an online and adaptive query selection algorithm, MARICH. MARICH uses a publicly available query dataset different from the private dataset. We experimentally show that the models extracted by MARICH achieve $56 - 86\%$ accuracy on the private dataset while using 959 - 8,429 queries. For both text and image data, we demonstrate that the models extracted by MARICH act as informative replicas of the target models and also yield high-fidelity replicas of the targets' prediction distributions. Typically, the functional equivalence attacks require model-specific techniques, while MARICH is model-oblivious while performing distributional equivalence attack. This poses an open question: is distributional equivalence extraction 'easier' than functional equivalence extraction, which is NP-hard [JCB+20]?

**Acknowledgments**

P. Karmakar acknowledges supports of the programme DesCartes and the National Research Foundation, Prime Minister's Office, Singapore under its Campus for Research Excellence and Technological Enterprise (CREATE) programme. D. Basu acknowledges the Inria-Kyoto University Associate Team "RELIANT", and the ANR JCJC for the REPUBLIC project (ANR-22-CE23-0003-01) for support. We also thank Cyriaque Rousselot for the interesting initial discussions.

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

# Appendix

## Table of Contents

## Broader impact

In this paper, we design a model extraction attack algorithm, MARICH, that aims to construct a model that has similar predictive distribution as that of a target model. In this direction, we show that popular deep Neural Network (NN) models can be replicated with a few number of queries and only outputs from their predictive API. We also show that this can be further used to conduct membership inference about the private training data that the adversary has no access to. Thus, MARICH points our attention to the vulnerabilities of the popular deep NN models to preserve the privacy of the users, whose data is used to train the deep NN models. Though every attack algorithm can be used adversarially, our goal is not to promote any such adversarial use.

Rather, in the similar spirit as that of the attacks developed in cryptography to help us to design better defenses and to understand vulnerabilities of the computing systems better, we conduct this research to understand the extent of information leakage done by an ML model under modest assumptions. We recommend it to be further used and studied for developing better privacy defenses and adversarial attack detection algorithms.

## A  Complete pseudocode of MARICH

---

**Algorithm 2** MARICH

---

**Input**: Target model: $f^T$, Query dataset: $\mathbf{D}^Q$, #Classes: $k$
**Parameter**: #initial samples: $n_0$, Training epochs: $E_{max}$, #Batches of queries: $T$, Query budget: $B$, Subsampling ratios: $\gamma_1, \gamma_2 \in (0,1]$
**Output**: Extracted model $f^E$

  1: //* Initialisation of the extracted model*//               ▷ *Phase 1*
  2: $Q_0^{train} \leftarrow n_0$ datapoints randomly chosen from $D^Q$
  3: $Y_0^{train} \leftarrow f^T(Q_0^{train})$               ▷ *Query the target model $f^T$ with $Q_0^{train}$*
  4: **for** epoch $\leftarrow 1$ to $E_{max}$ **do**
  5:      $f_0^E \leftarrow$ Train $f^E$ with $(Q_0^{train}, Y_0^{train})$
  6: **end for**
  7: //* Adaptive query selection to build the extracted model*//          ▷ *Phase 2*
  8: **for** $t \leftarrow 1$ to $T$ **do**
  9:      $Q_t^{entropy} \leftarrow$ ENTROPYSAMPLING$(f_{t-1}^E, \mathbf{D}^Q \setminus Q_{t-1}^{train}, B)$
10:      $Q_t^{grad} \leftarrow$ GRADIENTSAMPLING$(f_{t-1}^E, Q_t^{entropy}, \gamma_1 B)$
11:      $Q_t^{loss} \leftarrow$ LOSSSAMPLING$(f_{t-1}^E, Q_t^{grad}, Q_{t-1}^{train}, Y_{t-1}^{train}, \gamma_1 \gamma_2 B)$
12:      $Y_t^{new} \leftarrow f^T(Q_t^{loss})$           ▷ *Query the target model $f^T$ with $Q_t^{loss}$*
13:      $Q_t^{train} \leftarrow Q_{t-1}^{train} \cup Q_t^{loss}$
14:      $Y_t^{train} \leftarrow Y_{t-1}^{train} \cup Y_t^{new}$
15:      **for** epoch $\leftarrow 1$ to $E_{max}$ **do**
16:         $f_t^E \leftarrow$ Train $f_{t-1}^E$ with $(Q_t^{train}, Y_t^{train})$
17:      **end for**
18: **end for**
19: **return** Extracted model $f^E \leftarrow f_T^E$
20:
21: **EntropySampling** (extracted model: $f^E$, input data points: $X_{in}$, budget: $B$)
22: $Q_{entropy} \leftarrow \arg\max_{X \subset X_{in}, |X|=B} H(f^E(X_{in}))$
                                   ▷ *Select $B$ points with maximum entropy*
23: **return** $Q_{entropy}$
24:
25: **GradientSampling** (extracted model: $f^E$, input data points: $X_{in}$, budget: $\gamma_1 B$)
26: $E \leftarrow H(f^E(X_{in}))$
27: $G \leftarrow \{\nabla_x E \mid x \in X_{in}\}$
28: $C_{in} \leftarrow k$ centres of $G$ computed using K-means
29: $Q_{grad} \leftarrow \arg\min_{X \subset X_{in}, |X|=\gamma_1 B} \sum_{x_i \in X} \sum_{x_j \in C_{in}} \|\nabla_{x_i} E - \nabla_{x_j} E\|_2^2$
                     ▷ *Select $\gamma_1 B$ points from $X_{in}$ whose $\frac{\partial E}{\partial x}$ are closest to that of $C_{in}$*
30: **return** $Q_{grad}$
31:
32: **LossSampling** (extracted model: $f^E$, input data points: $X_{in}$, previous queries: $Q_{train}$, previous predictions: $Y_{train}$, budget: $\gamma_1 \gamma_2 B$)
33: $L \leftarrow l(Y_{train}, f^E(Q_{train}))$             ▷ *Compute the mismatch vector*
34: $Q_{mis} \leftarrow$ ARGMAXSORT$(L, k)$             ▷ *Select top-$k$ mismatching points*
35: $Q_{loss} \leftarrow \arg\min_{X \subset X_{in}, |X|=\gamma_1 \gamma_2 B} \sum_{x_i \in X} \sum_{x_j \in Q_{mis}} \|x_i - x_j\|_2^2$
                                   ▷ *Select $\gamma_1 \gamma_2 B$ points closest to $Q_{mis}$*
36: **return** $Q_{loss}$

---

# B Theoretical analysis: Proofs of Section 3

In this section, we elaborate the proofs for the Theorems 3.2 and 3.4.[3]

***Theorem 3.2*** (Upper Bounding Distributional Closeness). If we choose KL-divergence as the divergence function $D$, we can show that

$$D_{\mathrm{KL}}(\Pr(f^T_{\theta*}(Q), Q) \| \Pr(f^E_{\omega^*_{\mathrm{DEq}}}(Q), Q)) \leq \min_\omega \mathbb{E}_Q[l(f^T_{\theta*}(Q), f^E_\omega(Q))] - H(f^E_\omega(Q)).$$

*Proof.* Let us consider a query generating distribution $\mathcal{D}^Q$ on $\mathbb{R}^d$. A target model $f^T_{\theta*} : \mathbb{R}^d \to \mathcal{Y}$ induces a joint distribution over the query and the output (or label) space, denoted by $\Pr(f^T_{\theta*}, Q)$. Similarly, the extracted model $f^E_{\theta*} : \mathbb{R}^d \to \mathcal{Y}$ also induces a joint distribution over the query and the output (or label) space, denoted by $\Pr(f^E_\omega, Q)$.

$$D_{\mathrm{KL}}(\Pr(f^T_{\theta*}(Q), Q) \| \Pr(f^E_\omega(Q), Q))$$

$$= \int_{Q \in \mathbb{R}^d} \mathrm{d}\Pr(f^T_{\theta*}(Q), Q) \log \frac{\Pr(f^T_{\theta*}(Q), Q)}{\Pr(f^E_\omega(Q), Q)}$$

$$= \int_{Q \in \mathbb{R}^d} \Pr(f^T_{\theta*}(Q)|Q = q) \Pr(Q = q) \log \frac{\Pr(f^T_{\theta*}(Q)|Q = q)}{\Pr(f^E_\omega(Q)|Q = q)} \, \mathrm{d}q$$

$$= \int_{Q \in \mathbb{R}^d} \Pr(f^T_{\theta*}(Q)|Q = q) \Pr(Q = q) \log \Pr(f^T_{\theta*}(Q)|Q = q) \, \mathrm{d}q$$

$$- \int_{Q \in \mathbb{R}^d} \Pr(f^T_{\theta*}(Q)|Q = q) \Pr(Q = q) \log \Pr(f^E_\omega(Q)|Q = q) \, \mathrm{d}q$$

$$= \int_{Q \in \mathbb{R}^d} \Pr(f^T_{\theta*}(Q)|Q = q) \Pr(Q = q) \log \Pr(f^T_{\theta*}(Q)|Q = q) \, \mathrm{d}q + \mathbb{E}_{q \sim \mathcal{D}^Q} \left[ l(f^T_{\theta*}(q)), f^E_\omega(q)) \right]$$

$$\leq -H(f^T_{\theta*}(Q) \, \mathrm{d}q + \mathbb{E}_{q \sim \mathcal{D}^Q} \left[ l(f^T_{\theta*}(q)), f^E_\omega(q)) \right]$$

$$\leq -H(f^E_\omega(Q) \, \mathrm{d}q + \mathbb{E}_{q \sim \mathcal{D}^Q} \left[ l(f^T_{\theta*}(q)), f^E_\omega(q)) \right] \tag{8}$$

The last inequality holds true as the extracted model $f^E_\omega$ is trained using the outputs of the target model $f^T_{\theta*}$. Thus, by data-processing inequality, its output distribution possesses less information than that of the target model. Specifically, we know that if $Y = f(X)$, $H(Y) \leq H(X)$.

Now, by taking $\min_\omega$ on both sides, we obtain

$$D_{\mathrm{KL}}(\Pr(f^T_{\theta*}(Q), Q) \| \Pr(f^E_{\omega^*_{\mathrm{DEq}}}(Q), Q)) \leq \min_\omega \mathbb{E}_Q[l(f^T_{\theta*}(Q), f^E_\omega(Q))] - H(f^E_\omega(Q)).$$

Here, $\omega^*_{\mathrm{DEq}} \triangleq \arg\min_\omega D_{\mathrm{KL}}(\Pr(f^T_{\theta*}(Q), Q) \| \Pr(f^E_\omega(Q), Q))$. The equality exists if minima of LHS and RHS coincide. $\square$

***Theorem 3.4*** (Lower Bounding Information Leakage). The information leaked by any Max-Information attack (Equation 3) is lower bounded as follows:

$$I(\Pr(f^T_{\theta*}(Q), Q) \| \Pr(f^E_{\omega^*_{\mathrm{MaxInf}}}(Q), Q)) \geq \max_\omega -\mathbb{E}_Q[l(f^T_{\theta*}(Q), f^E_\omega(Q))] + H(f^E_\omega(Q)).$$

*Proof.* Let us consider the same terminology as the previous proof. Then,

$$I(\Pr(f^T_{\theta*}(Q), Q) \| \Pr(f^E_\omega(Q), Q))$$

$$= H(f^T_{\theta*}(Q), Q) + H(f^E_\omega(Q), Q) - H(f^T_{\theta*}(Q), f^E_\omega(Q), Q)$$

$$= H(f^T_{\theta*}(Q), Q) + H(f^E_\omega(Q), Q) - H(f^E_\omega(Q), Q|f^T_{\theta*}(Q)) + H(f^T_{\theta*}(Q))$$

$$\geq H(f^E_\omega(Q), Q) - H(f^E_\omega(Q), Q|f^T_{\theta*}(Q)) \tag{9}$$

$$\geq H(f^E_\omega(Q)) - H(f^E_\omega(Q), Q|f^T_{\theta*}(Q)) \tag{10}$$

$$\geq H(f^E_\omega(Q)) - \mathbb{E}_Q[l(f^E_\omega(Q), f^T_{\theta*}(Q))] \tag{11}$$

---

[3]Throughout the proofs, we slightly abuse the notation to write $l(\Pr(X), \Pr(Y))$ as $l(X, Y)$ for avoiding cumbersome equations.

The inequality of Equation 9 is due to the fact that entropy is always non-negative. Equation 10 holds true as $H(X,Y) \geq \max\{H(X), H(Y)\}$ for two random variables $X$ and $Y$. The last inequality is due to the fact that conditional entropy of two random variables $X$ and $Y$, i.e. $H(X|Y)$, is smaller than or equal to their cross entropy, i.e. $l(X,Y)$ (Lemma B.1).

Now, by taking $\max_\omega$ on both sides, we obtain

$$I(\Pr(f_{\theta^*}^T(Q), Q) \| \Pr(f_{\omega_{\text{MaxInf}}^*}^E(Q), Q)) \leq \max_\omega -\mathbb{E}_Q[l(f_{\theta^*}^T(Q), f_\omega^E(Q))] + H(f_\omega^E(Q)).$$

Here, $\omega_{\text{MaxInf}}^* \triangleq \arg\max_\omega I(\Pr(f_{\theta^*}^T(Q), Q) \| \Pr(f_{\omega_{\text{MaxInf}}^*}^E(Q), Q))$. The equality exists if maxima of LHS and RHS coincide. $\square$

**Lemma B.1** (Relating Cross Entropy and Conditional Entropy). *Given two random variables $X$ and $Y$, conditional entropy*

$$H(X|Y) \leq l(X,Y). \tag{12}$$

*Proof.* Here, $H(X|Y) \triangleq -\int \Pr(x,y) \log \frac{\Pr(x,y)}{\Pr(y)} d\nu_1(X) d\nu_2(Y)$ and $l(X,Y) \triangleq l(\Pr(X), \Pr(Y)) = -\int \Pr(x) \ln \Pr(y) d\nu_1(X) d\nu_2(Y)$ denotes the cross-entropy, given reference measures $\nu_1$ and $\nu_2$.

$$\begin{aligned}
l(X,Y) &= H(X) + D_{\text{KL}}(\Pr(X) \| \Pr(Y)) \\
&= H(X|Y) + I(X;Y) + D_{\text{KL}}(P_X \| P_Y) \\
&\geq H(X|Y)
\end{aligned}$$

The last inequality holds as both mutual information $I$ and KL-divergence $D_{\text{KL}}$ are non-negative functions for any $X$ and $Y$. $\square$

## C   A review of active sampling strategies

**Least Confidence sampling (LC).**   Least confidence sampling method [Set09, LS06] iteratively selects the subset of $k$ data points from a data pool, which are most uncertain at that particular instant. The uncertainty function $(u(\cdot|f_\omega) : \mathcal{X} \to [0, 1])$ is defined as

$$u(x|f_\omega) \triangleq 1 - \Pr(\hat{y}|x),$$

where $\hat{y}$ is the predicted class by a model $f_\omega$ for input $x$.

**Margin Sampling (MS).**   In margin sampling [JG19], a subset of $k$ points is selected from a data pool, such that the subset demonstrates the minimum margin, where $\text{margin}(\cdot|f_\omega) : \mathcal{X} \to [0, 1]$ is defined as

$$\text{margin}(x|f_\omega) \triangleq \Pr(\widehat{y_1}(x)|x, f_\omega) - \Pr(\widehat{y_2}(x)|x, f_\omega),$$

where $f_\omega$ is the model, and $\hat{y}_1(x)$ and $\hat{y}_2(x)$ are respectively the highest and the second highest scoring classes returned by $f_\omega$.

**Entropy Sampling (ES).**   Entropy sampling, also known as uncertainty sampling [LG94], iteratively selects a subset of $k$ datapoints with the highest uncertainty from a data pool. The uncertainty is defined by the entropy function of the prediction vector, and is computed using all the probabilities returned by the model $f_\omega$ for a datapoint $x$. For a given point $x$ and a model $f_\omega$, entropy is defined as

$$\text{entropy}(x|f_\omega) \triangleq - \sum_{a=1}^{|\mathcal{Y}|} p_a \log(p_a),$$

where $p_a = \Pr[f_\omega(x) = a]$ for any output class $a \in \{1, \ldots, |\mathcal{Y}|\}$. [LG94] mention that while using this strategy, "*the initial classifier plays an important role, since without it there may be a long period of random sampling before examples of a low frequency class are stumbled upon*". This is similar to our experimental observation that ES often demonstrate high variance in its outcomes.

**Core-set Algorithms.**   Here, we discuss some other interesting works in active learning using core-sets, and the issues to directly implement them in our problem.

[KZCI21] aims to identify a subset of training dataset for training a specific model. The algorithm needs *white-box access to the model*, and also needs to do a forward pass over the whole training dataset. A white-box sampling algorithm and the assumption of being able to retrieve predictions over the whole training dataset are not feasible in our problem setup. Relaxing the white-box access, [KDR$^+$21] proposes to us the average loss over the training dataset or a significantly diverse validation set. Then, the gradient of loss on this training or validation set is compared with that of the selected mini-batch of data points. In a black-box attack, we do not have access to average loss over a whole training or validation dataset. Thus, it is not feasible to deploy the proposed algorithm.

Following another approach, [MBL20] proposes an elegant pre-processing algorithm to select a core-set of a training dataset. This selection further leads to an efficient incremental gradient based training methodology. But in our case, we sequentially query the black-box model to obtain a label for a query point and use them further for training the extracted model. Thus, creating a dataset beforehand and using them further to pre-process will not lead to an adaptive attack and also will not reduce the query budget. Thus, it is out of scope of our work. [KS22] deploys an auxiliary classifier to first use a subset of labelled datapoints to create low dimensional embeddings. Then according to the query budget, it chooses the points from the sparse regions from each cluster found from the low dimensional embeddings. This is another variant of uncertainty sampling that hypothesises the points from the sparse region are more informative in terms of loss and prediction entropy. This design technique is at the same time model specific, and thus incompatible to our interest. Additionally, this approach also increases the requirement of queries to the target model.

**K-Center sampling (KC).**   Thus, we focus on a version of core-set algorithm, namely K-Center sampling, that is applicable in our context. K-Center sampling as an active learning algorithm that has been originally proposed to train CNNs sample-efiiciently [SS18]. For a given data pool, K-Center sampling iteratively selects the $k$ datapoints that minimise the core set loss (see Equation 3, [SS18]) the most for a given model $f_\omega$. In this method, the embeddings (from the model under training) of the data points are used as the representative vectors, and K-Center algorithm is applied on these representative vectors.

**Random Sampling (RS).** In random sampling, a subset of $k$ datapoints are selected from a data pool uniformly at random.

In our experiments, for query selection at time $t$, the extracted model at time $t-1$, i.e. $f_{t-1}^E$, is used as $f_\omega$, and data pool at time $t$ is the corresponding query dataset, except the datapoints that has been selected before step $t$. Hereafter, we deploy a modified version of the framework developed by [Hua21] to run our experiments with the aforementioned active learning algorithms.

# D  Extended experimental analysis

In this section, we step-wise elaborate further experimental setups and results that we skipped for the brevity of space in the main draft. Specifically, we conduct our experiments in six experimental setups. Each experimental setup corresponds to a triplet (target model architecture trained on a private dataset, extracted model architecture, query dataset). Here, we list these six experimental setups in detail

1. A Logistic Regression (LR) model trained on MNIST, a LR model for extraction, EMNIST dataset for querying

2. A Logistic Regression (LR) model trained on MNIST, a LR model for extraction, CIFAR10 dataset for querying

3. A CNN model trained on MNIST, a CNN model for extraction, EMNIST dataset for querying

4. A ResNet model trained on CIFAR10, a CNN model for extraction, ImageNet dataset for querying

5. A ResNet model trained on CIFAR10, a ResNet18[4] model for extraction, ImageNet dataset for querying

6. A BERT model[5] trained on BBCNews, a BERT model for extraction, AGNews dataset for querying

For each of the experimental setups, we evaluate five types of performance evaluations, which are elaborated in Section D.1, D.2.1, D.2.2, D.3, and D.4. While each of the following sections contain illustrations of the different performance metrics evaluating efficacy of the attack and corresponding discussions, Table 2- 4 contain summary of all queries used, accuracy, and membership inference statistics for all the experiments.

## D.1  Test accuracy of extracted models

Test accuracy of the extracted model and its comparison with the test accuracy of the target model on a subset of the private training dataset, which was used by neither of these models, is the most common performance metric used to evaluate the goodness of the attack algorithm. The attacks designed solely to optimise this performance metric are called the task accuracy model extraction attacks [JCB+20].

With MARICH, we aim to extract models that have prediction distributions closest to that of the target model. Our hypothesis is constructing such a prediction distribution lead to a model that also has high accuracy on the private test dataset, since accuracy is a functional property of the prediction distribution induced by a classifier. In order to validate this hypothesis, we compute test accuracies of the target models, and models extracted by MARICH and other active sampling algorithms in six experimental setups. We illustrate the evolution curves of accuracies over increasing number of queries in Figure 5.

To compare MARICH with other active learning algorithms, we attack the same target models using K-centre sampling, Least Confidence sampling, Margin Sampling, Entropy Sampling, and Random Sampling algorithms (ref. Appendix C) using the same number of queries as used for MARICH in each setup.

From Figure 5, we observe that **in most of the cases MARICH outperforms all other competing algorithms**.

In this process, MARICH uses $\sim 500 - 8000$ queries, which is a small fraction of the corresponding query datasets. This also indicates towards the query-efficiency of MARICH.

**Extraction of a ResNet trained on CIFAR10 with a ResNet18.**  Along with the five experimental setups mentioned in the paper, we trained a ResNet with CIFAR10 dataset ($\mathbf{D}^T$ here), that shows a test accuracy of $91.82\%$ on a disjoint test set. We use ImageNet as $\mathbf{D}^Q$ here, to extract a ResNet18 model from the target model. We have restrained from discussing this setup in the main paper due to brevity of space.

---

[4]We begin with a pre-trained ResNet18 model from `https://pytorch.org/vision/main/models/generated/torchvision.models.resnet18.html`

[5]We use the pre-trained BERT model from `https://huggingface.co/bert-base-cased`

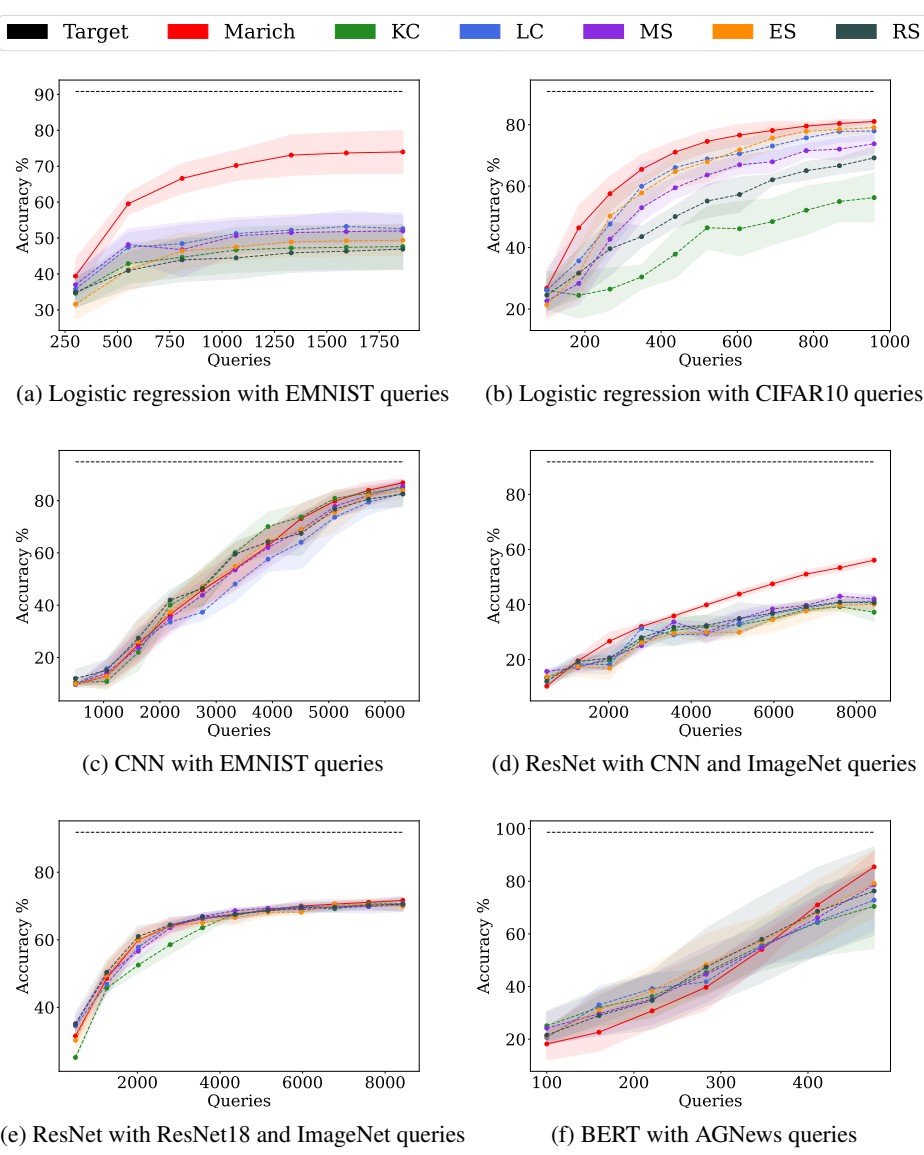

(a) Logistic regression with EMNIST queries

(b) Logistic regression with CIFAR10 queries

(c) CNN with EMNIST queries

(d) ResNet with CNN and ImageNet queries

(e) ResNet with ResNet18 and ImageNet queries

(f) BERT with AGNews queries

Figure 5: Comparison of test accuracies achieved by models extracted by different active sampling algorithms.

MARICH extracts a ResNet18 using $8429$ queries, and the extracted ResNet18 shows test accuracy of $71.65 \pm 0.88\%$. On the other hand, models extracted using Best of Competitors (BoC) using ImageNet queries shows accuracy of $70.67 \pm 0.12\%$.

### D.2   Fidelity of the prediction distributions of The extracted models

Driven by the distributional equivalence extraction principle, the central goal of MARICH is to construct extracted models whose prediction distributions are closest to the prediction distributions of corresponding target models. From this perspective, in this section, we study the fidelity of the prediction distributions of models extracted by MARICH and other active sampling algorithms, namely K-centre sampling, Least Confidence sampling, Margin Sampling, Entropy Sampling, and Random Sampling.

#### D.2.1   KL-divergence between prediction distributions

First, as the metric of distributional equivalence, we evaluate the KL-divergence between the prediction distributions of the models extracted by MARICH and other active sampling algorithms. In Figure 6, we report the box-plot (mean, median $\pm$ 25 percentiles) of KL-divergences (in log-scale) calculated from 5 runs for each of 10 models extracted by each of the algorithms.

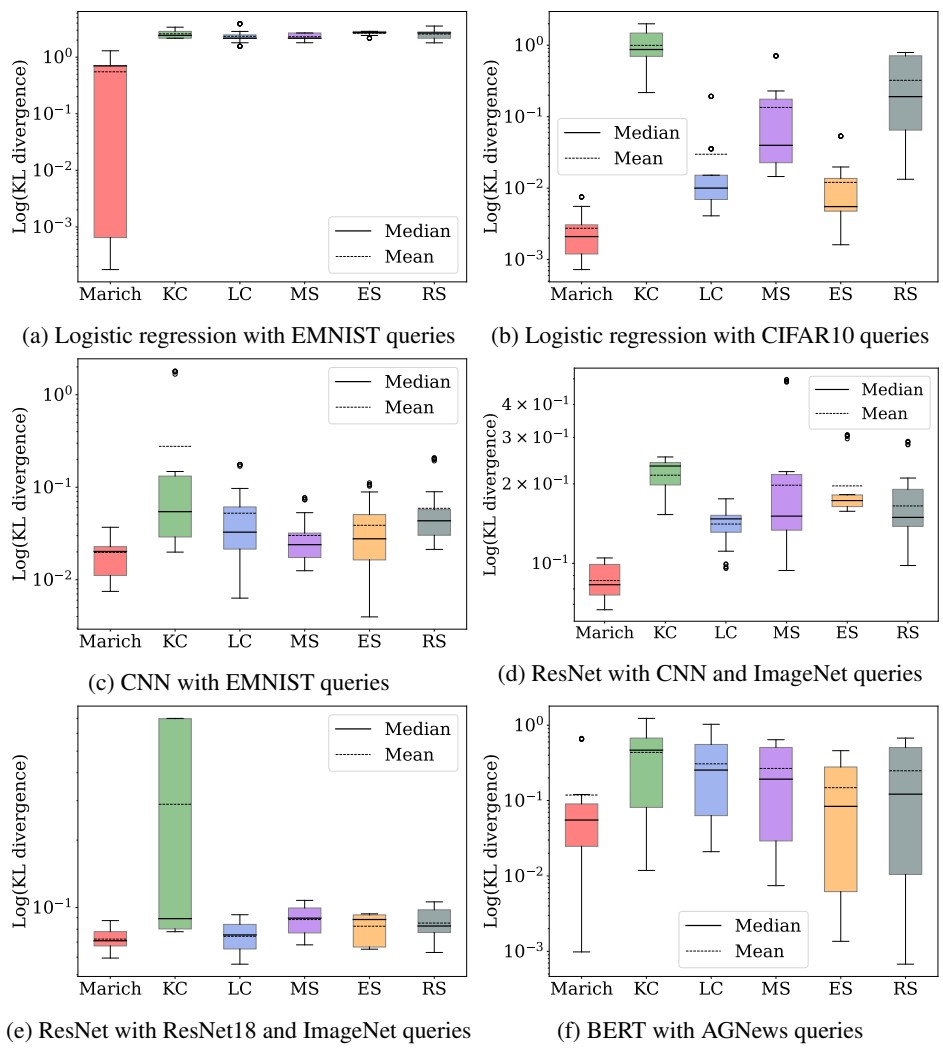

Figure 6: Comparison of KL-divergences (in log-scale) between the target prediction distributions and prediction distributions of the models extracted by different active learning algorithms.

**Results.** Figure 6 shows that the KL-divergence achieved by the prediction distributions of models extracted using MARICH are at least $\sim 2 - 10$ times less than that of the other competing algorithms. This validates our claim that MARICH *yields distributionally closer extracted model $f^E$ from the target model $f^T$ than existing active sampling algorithms.*

### D.2.2 Prediction agreement

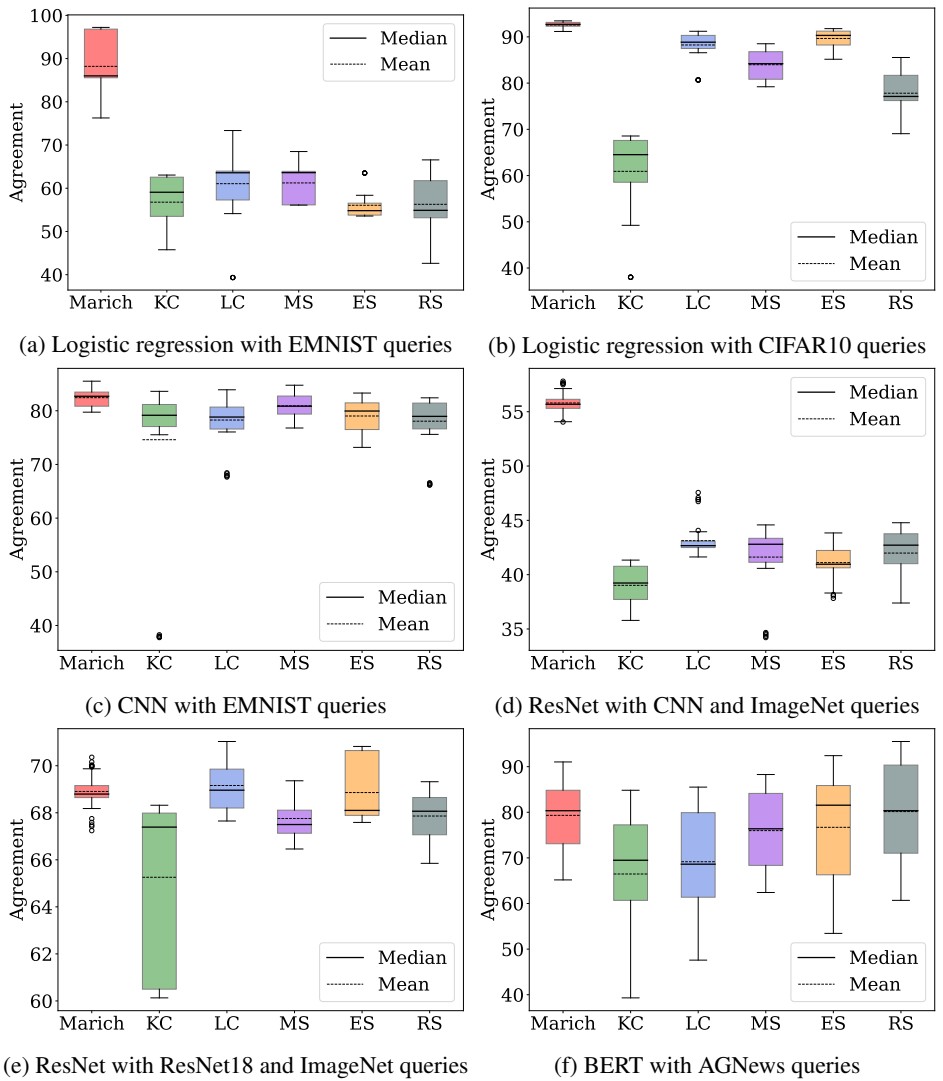

(a) Logistic regression with EMNIST queries  (b) Logistic regression with CIFAR10 queries

(c) CNN with EMNIST queries  (d) ResNet with CNN and ImageNet queries

(e) ResNet with ResNet18 and ImageNet queries  (f) BERT with AGNews queries

Figure 7: Comparison of agreement in predictions (in %) between the target model and the models extracted by different active learning algorithms.

In Figure 7, we illustrate the agreement in predictions of $f^E$ with $f^T$ on test datasets using different active learning algorithms. Prediction agreement functions as another metric of fidelity of prediction distributions constructed by extracted models in comparison with those of the target models.

Similar to Figure 6, we report the box-plot (mean, median $\pm$ 25 percentiles) of prediction agreements (in %) calculated from 5 runs for each of 10 models extracted by each of the algorithms.

**Results.** We observe that the prediction distributions extracted by MARICH achieve almost same to $\sim 30\%$ higher prediction agreement in comparison with the competing algorithms. Thus, we infer that in this particular case MARICH *achieves better fidelity than the other active sampling algorithms, in some instances, while it is similar to the BoC in some instances.*

### D.3 Fidelity of parameters of the extracted models

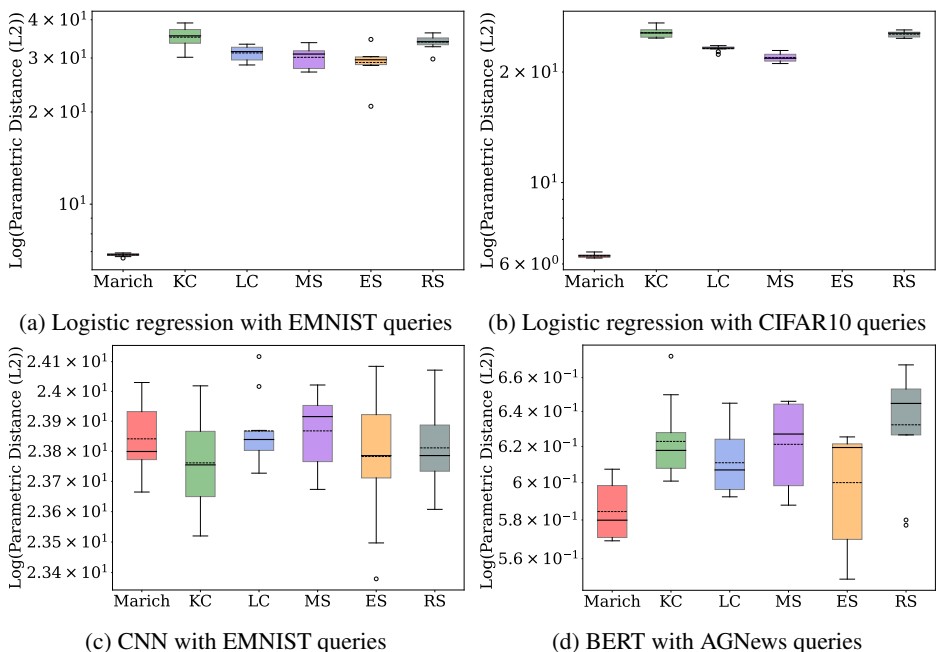

(a) Logistic regression with EMNIST queries     (b) Logistic regression with CIFAR10 queries

(c) CNN with EMNIST queries     (d) BERT with AGNews queries

Figure 8: Comparison of parametric fidelity for MARICH and different active learning algorithms.

Generating extracted models with low parametric fidelity is not a main goal or basis of the design principle of MARICH. Since parametric fidelity is a popularly studied metric to evaluate goodness of model extraction, in Figure 8, we depict the parametric fidelity of models extracted by different active learning algorithms.

Let $\mathbf{w}_E$ be the parameters of the extracted model and $\mathbf{w}_T$ be the parameters of the target model. We define parametric fidelity as $F_{\mathbf{w}} \triangleq \log \|\mathbf{w}_E - \mathbf{w}_T\|_2$. Since the parametric fidelity is only computable when the target and extracted models share the same architecture, we report the four instances here where MARICH is deployed with the same architecture as that of the target model. For logistic regression, we compare all the weights of the target and the extracted models. For BERT and CNN, we compare between the weights in the last layers of these models.

**Results.** For LR, we observe that the LR models extracted by MARICH have $20 - 30$ times lower parametric fidelity than the extracted LR models of the competing algorithms. For BERT, the BERT extracted by MARICH achieves $0.4$ times lower parametric fidelity than the Best of Competitors (BoC). As an exception, for CNN, the model extracted by K-center sampling achieves $0.996$ times less parametric fidelity than that of MARICH.

Thus, we conclude that MARICH *as a by-product of its distributionally equivalent extraction principle also extracts model with high parametric fidelity*, which is often better than the competing active sampling algorithms.

## D.4   Membership inference with the extracted models

A main goal of MARICH is to conduct a Max-Information attack on the target model, i.e. to extract an informative replica of its predictive distribution that retains the most information about the private training dataset. Due to lack of any direct measure of informativeness of an extracted model with respect to a target model, we run Membership Inference (MI) attacks using the models extracted by MARICH, and other competing active sampling algorithms. High accuracy and agreement in MI attacks conducted on extracted models of MARICH and the target models implicitly validate our claim that MARICH is able to conduct a Max-Information attack.

**Observation 1.** From Figure 9, we see that in most cases the probability densities of the membership inference are closer to the target model when the model is extracted using MARICH, than using all other active sampling algorithms (BoC, Best of Competitors).

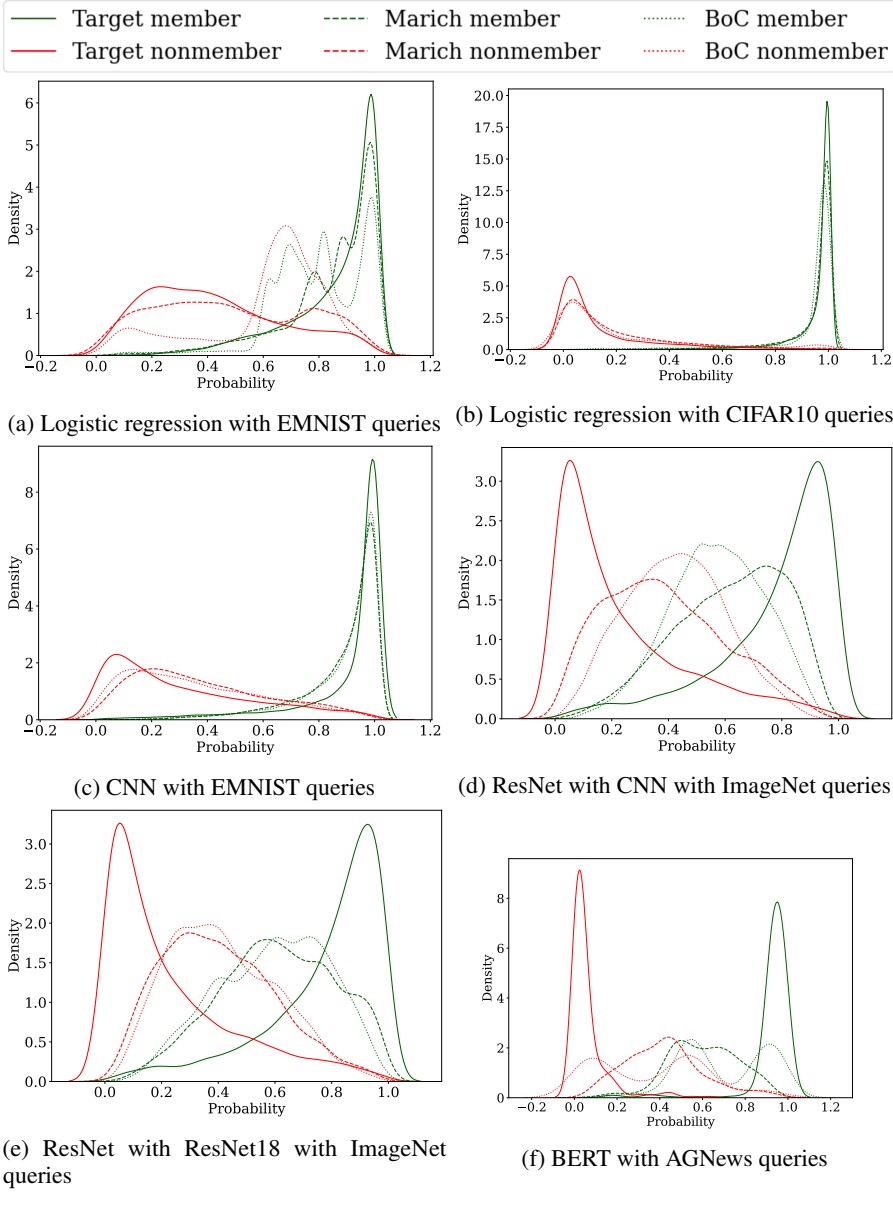

Figure 9: Comparison among membership vs. non-membership probability densities for membership attacks against models extracted by MARICH, the best of competitors (BoC) and the target model. Each figure represents the model class and query dataset. Memberships and non-memberships inferred from the model extracted by MARICH are significantly closer to the target model.

**Observation 2.** In Figure 10, we present the agreements from the member points, nonmember points and overall agreement curves for varying membership thresholds, along with the AUCs of the overall membership agreements. We see that in most cases, the agreement curves for the models extracted using MARICH are above those for the models extracted using random sampling, thus AUCs are higher for the models extracted using MARICH.

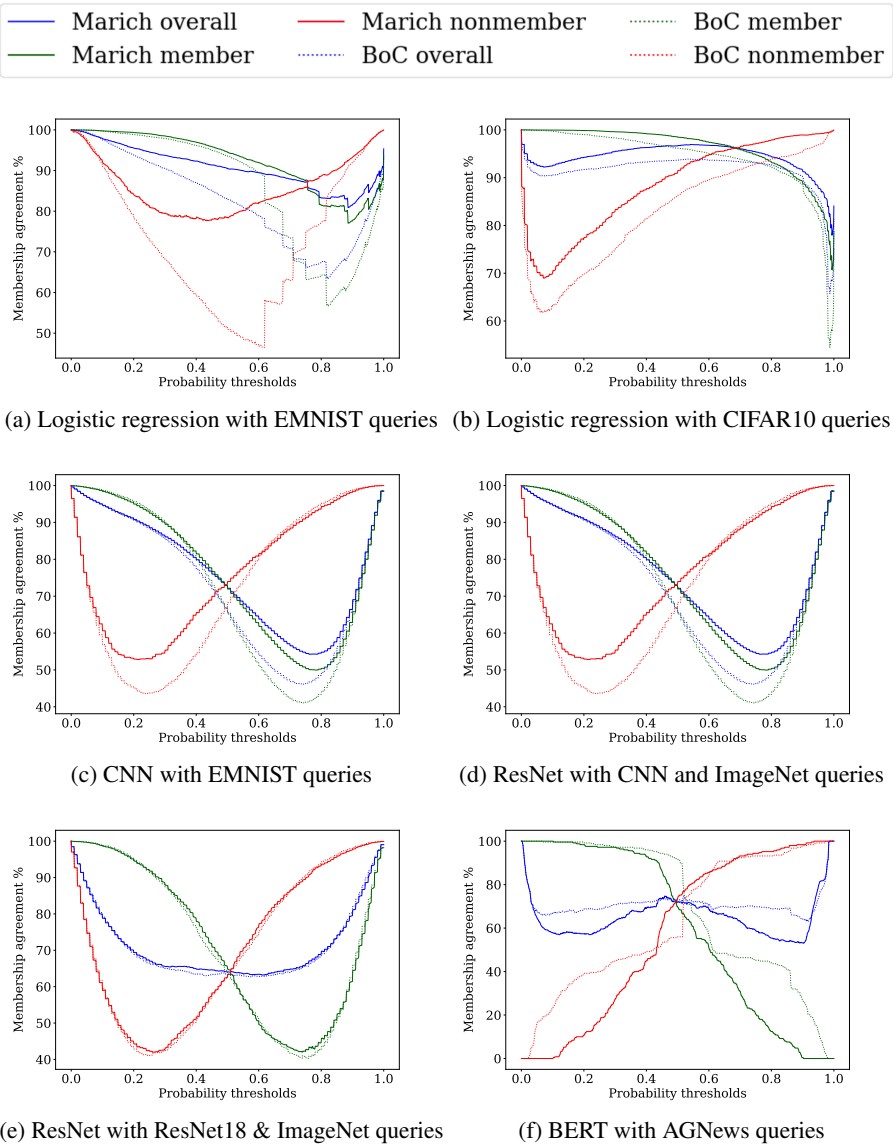

(a) Logistic regression with EMNIST queries  (b) Logistic regression with CIFAR10 queries

(c) CNN with EMNIST queries  (d) ResNet with CNN and ImageNet queries

(e) ResNet with ResNet18 & ImageNet queries  (f) BERT with AGNews queries

Figure 10: Comparison of membership, nonmembership and overall agreements of membership attacks against models extracted by MARICH and the best of competitors with the target model trained with MNIST. Each figure represents the model class and query dataset. Membership agreement of the models extracted by MARICH are higher.

**Observation 3.** In Table 2, 3, and 4, we summarise the MI accuracy on the private training dataset, Nonmembership inference accuracy on the private training dataset, Agreement in MI w.r.t. the MI on the target model, and AUC of Agreement in MI with that of the target model for Logistic Regression (LR), CNN, ResNet, and BERT target models. We observe that, while compared with other active sampling algorithms, out of 6 combinations of (target model, extracted model, query dataset) under study, the models extracted by MARICH achieve the highest accuracy in MI and agreement in MI w.r.t. the target model, in most of the instances.

**Results.** *These observations support our claim that model extraction using* MARICH *gives models are accurate and informative replica of the target model.*

Table 2: Model extraction and membership inference statistics for extracting a Logistic Regression model and a CNN

| Member Dataset | Target Model | Attack Dataset | Algorithm used | Non-member Dataset | #Queries | Membership acc | Nonmembership acc | Overall membership acc | Overall membership agreement | Membership agreement AUC | Accuracy |
|---|---|---|---|---|---|---|---|---|---|---|---|
| MNIST | LR | - | - | EMNIST | - | 95.37% | 65.84% | 87.99% | - | - | 90.82% |
| MNIST | LR | EMNIST | Marich | EMNIST | 1863 | 93.43% | 57.58% | 84.47% | 90.34% | 90.89% | 73.98±5.96% |
| MNIST | LR | EMNIST | ES | EMNIST | 1863 | 97.22% | 15.88% | 75.88% | 84.39% | 82.32% | 49.37±4.24% |
| MNIST | LR | EMNIST | KC | EMNIST | 1863 | 97.80% | 14.56% | 76.99% | 87.84% | 82.30% | 47.62±6.25% |
| MNIST | LR | EMNIST | MS | EMNIST | 1863 | 97.63% | 18.42% | 77.83% | 86.49% | 82.74% | 52.60±3.55% |
| MNIST | LR | EMNIST | LC | EMNIST | 1863 | 95.64% | 25.10% | 78.00% | 80.11% | 83.07% | 51.96±5.05% |
| MNIST | LR | EMNIST | RS | EMNIST | 1863 | 98.23% | 11.92% | 76.65% | 88.08% | 82.26% | 46.91±5.69% |
| MNIST | LR | - | - | CIFAR10 | - | 99.11% | 94.76% | 98.02% | - | - | 90.82% |
| MNIST | LR | CIFAR10 | Marich | CIFAR10 | 959 | 97.72% | 92.14% | 96.32% | 96.89% | 94.32% | 81.06±0.56% |
| MNIST | LR | CIFAR10 | ES | CIFAR10 | 959 | 96.47% | 86.16% | 93.89% | 96.53% | 91.29% | 79.09±1.82% |
| MNIST | LR | CIFAR10 | KC | CIFAR10 | 959 | 95.87% | 81.64% | 92.31% | 92.32% | 89.64% | 56.26±7.97% |
| MNIST | LR | CIFAR10 | MS | CIFAR10 | 959 | 96.79% | 82.82% | 93.30% | 93.16% | 90.92% | 77.93±2.94% |
| MNIST | LR | CIFAR10 | LC | CIFAR10 | 959 | 96.19% | 86.25% | 93.71% | 93.67% | 91.53% | 73.78±2.95% |
| MNIST | LR | CIFAR10 | RS | CIFAR10 | 959 | 95.84% | 81.20% | 92.18% | 92.17% | 90.11% | 69.18±3.72% |
| MNIST | CNN | - | - | EMNIST | - | 93.71% | 78.76% | 89.97% | - | - | 94.83% |
| MNIST | CNN | EMNIST | Marich | EMNIST | 1863 | 94.55% | 78.86% | 90.62% | 87.27% | 86.72% | 86.83±1.62% |
| MNIST | CNN | EMNIST | ES | EMNIST | 1863 | 95.44% | 73.66% | 89.99% | 87.11% | 86.38% | 83.90±3.10% |
| MNIST | CNN | EMNIST | KC | EMNIST | 1863 | 95.35% | 74.48% | 90.13% | 87.49% | 85.94% | 84.94±2.44% |
| MNIST | CNN | EMNIST | MS | EMNIST | 1863 | 94.59% | 79.04% | 90.70% | 86.74% | 86.08% | 82.72±4.47% |
| MNIST | CNN | EMNIST | LC | EMNIST | 1863 | 94.39% | 76.30% | 89.86% | 86.75% | 86.10% | 85.53±2.45% |
| MNIST | CNN | EMNIST | RS | EMNIST | 1863 | 94.17% | 80.42% | 90.73% | 87.53% | 86.97% | 82.52±4.87% |

Table 3: Model extraction and membership inference statistics for extracting a ResNet

| Member Dataset | Target Model | Attack Model | Attack Dataset | Algorithm used | Non-member Dataset | #Queries | Membership acc | Nonmembership acc | Overall membership acc | Overall membership agreement | Membership agreement AUC | Accuracy |
|---|---|---|---|---|---|---|---|---|---|---|---|---|
| CIFAR10 | ResNet | - | - | - | ImageNet | - | 97.70% | 56.80% | 93.61% | - | - | 91.82% |
| CIFAR10 | ResNet | CNN | ImageNet | Marich | ImageNet | 8429 | 70.84% | 68.44% | 69.64% | 64.14% | 71.78% | **56.11±1.34%** |
| CIFAR10 | ResNet | CNN | ImageNet | ES | ImageNet | 8429 | 65.00% | 71.62% | 68.31% | 62.94% | 71.36% | 40.11±1.94% |
| CIFAR10 | ResNet | CNN | ImageNet | KC | ImageNet | 8429 | 65.80% | **71.98%** | 68.89% | 63.68% | 71.53% | 37.21±3.41% |
| CIFAR10 | ResNet | CNN | ImageNet | MS | ImageNet | 8429 | 68.48% | 71.96% | 70.22% | 64.48% | 71.97% | 41.27±0.85% |
| CIFAR10 | ResNet | CNN | ImageNet | LC | ImageNet | 8429 | 70.60% | 71.68% | **71.14%** | **65.47%** | **72.29%** | 42.05±1.38% |
| CIFAR10 | ResNet | CNN | ImageNet | RS | ImageNet | 8429 | 67.56% | 68.68% | 68.12% | 63.41% | 71.39% | 40.66±2.58% |
| CIFAR10 | ResNet | ResNet18 | ImageNet | Marich | ImageNet | 8429 | 99.27% | 10.54% | 90.40% | 93.84% | **76.51%** | **71.65±0.88%** |
| CIFAR10 | ResNet | ResNet18 | ImageNet | ES | ImageNet | 8429 | **100.00%** | 0.02% | 90.00% | **97.29%** | 72.33% | 69.92±1.54% |
| CIFAR10 | ResNet | ResNet18 | ImageNet | KC | ImageNet | 8429 | 99.79% | 2.06% | 90.02% | 94.06% | 72.40% | 70.67±0.12% |
| CIFAR10 | ResNet | ResNet18 | ImageNet | MS | ImageNet | 8429 | 99.99% | 0.28% | 90.02% | 96.28% | 72.81% | 70.66±2.29% |
| CIFAR10 | ResNet | ResNet18 | ImageNet | LC | ImageNet | 8429 | 99.99% | 0.04% | 90.00% | 96.77% | 71.94% | 70.26±1.44% |
| CIFAR10 | ResNet | ResNet18 | ImageNet | RS | ImageNet | 8429 | 99.94% | 1.40% | 90.08% | 95.41% | 72.94% | 70.49±2.02% |

Table 4: Model extraction and membership inference statistics for extracting a BERT model

| Member Dataset | Target Model | Attack Model | Attack Dataset | Algorithm used | Non-member Dataset | Queries | Membership acc | Nonmembership acc | Overall membership acc | Overall membership agreement | Membership agreement AUC | Accuracy |
|---|---|---|---|---|---|---|---|---|---|---|---|---|
| BBC News | BERT | - | - | - | AG News | - | **83.60%** | 66.80% | **75.20%** | - | 65.14% | 98.62% |
| BBC News | BERT | BERT | AG News | Marich | AG News | 474 | 82.80% | 48.40% | 65.60% | **74.80%** | 59.36% | **85.45±5.96%** |
| BBC News | BERT | BERT | AG News | ES | AG News | 474 | 68.40% | 55.60% | 62.00% | 66.00% | 61.10% | 79.25±12.67% |
| BBC News | BERT | BERT | AG News | KC | AG News | 474 | 62.00% | **72.80%** | 75.00% | 63.39% | **71.07%** | 70.45±15.94% |
| BBC News | BERT | BERT | AG News | MS | AG News | 474 | 77.20% | 43.20% | 61.60% | 62.40% | 57.93% | 72.82±11.59% |
| BBC News | BERT | BERT | AG News | LC | AG News | 474 | 80.00% | 47.60% | 62.80% | 74.80% | 74.80% | 78.65±7.22% |
| BBC News | BERT | BERT | AG News | RS | AG News | 474 | 78.00% | | | 63.80% | 58.43% | 76.31±16.78% |

# E    Significance and comparison of three sampling strategies

Given the bi-level optimization problem, we came up with MARICH in which three sampling methods are used in the order: (i) ENTROPYSAMPLING, (ii) ENTROPYGRADIENTSAMPLING, and (iii) LOSSSAMPLING.

These three sampling techniques contribute to different goals:

- ENTROPYSAMPLING selects points about which the classifier at a particular time step is most confused
- ENTROPYGRADIENTSAMPLING uses gradients of entropy of outputs of the extracted model w.r.t. the inputs as embeddings and selects points behaving most diversely at every time step.
- LOSSSAMPLING selects points which produce highest loss when loss is calculated between target model's output and extracted model's output.

One can argue that the order is immaterial for the optimization problem. But looking at the algorithm practically, we see that ENTROPYGRADIENTSAMPLING and LOSSSAMPLING incur much higher time complexity than ENTROPYSAMPLING. Thus, using ENTROPYSAMPLING on the entire query set is more efficient than the others. This makes us put ENTROPYSAMPLING as the first query selection strategy.

As per the optimization problem in Equation (7), we are supposed to find points that show highest mismatch between the target and the extracted models after choosing the query subset maximising the entropy. This leads us to the idea of LOSSSAMPLING. But as only focusing on loss between models may choose points from one particular region only, and thus, decreasing the diversity of the queries. We use ENTROPYGRADIENTSAMPLING before LOSSSAMPLING. This ensures selection of diverse points with high performance mismatch.

In Figure 11, we experimentally see the time complexities of the three components used. These are calculated by applying the sampling algorithms on a logistic regression model, on mentioned slices of MNIST dataset.

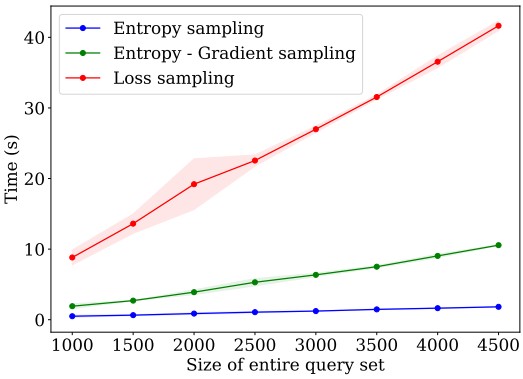

Figure 11: Runtime comparison of three sampling strategies to select queries from 4500 datapoints.

Table 5: Time complexity of different sampling Strategies

| Sampling Algorithm | Query space size | #Selected queries | Time (s) |
|---|---|---|---|
| Entropy Sampling | 4500 | 100 | $1.82 \pm 0.04$ |
| Entropy-Gradient Sampling | 4500 | 100 | $10.56 \pm 0.07$ |
| Loss Sampling | 4500 | 100 | $41.64 \pm 0.69$ |

# F    Performance against differentially private target models

In this section, we aim to verify performance of MARICH against privacy-preserving mechanisms. Specifically, we apply a $(\varepsilon, \delta)$-Differential Privacy (DP) inducing mechanism [DMNS06, DBB21] on the target model to protect the private training dataset. There are three types of methods to ensure DP: output perturbation [DMNS06], objective perturbation [CMS11, DBB18], and gradient perturbation [ACG+16]. Since output perturbation and gradient perturbation methods scale well for nonlinear deep networks, we focus on them as the defense mechanism against MARICH's queries.

**Gradient perturbation-based defenses.**   DP-SGD [ACG+16] is used to train the target model on the member dataset. This mechanism adds noise to the gradients and clip them while training the target model. We use the default implementation of Opacus [YSS+21] to conduct the training in PyTorch.

Following that, we attack the $(\varepsilon, \delta)$-DP target models using MARICH and compute the corresponding accuracy of the extracted models. In Figure 12, we show the effect of different privacy levels $\varepsilon$ on the achieved accuracy of the extracted Logistic Regression model trained with MNIST dataset and queried with EMNIST dataset. Specifically, we assign $\delta = 10^{-5}$ and vary $\varepsilon$ in $\{0.2, 0.5, 1, 2, \infty\}$. Here, $\varepsilon = \infty$ corresponds to the model extracted from the non-private target model.

We observe that the accuracy of the models extracted from private target models are approximately $2.3 - 7.4\%$ lower than the model extracted from the non-private target model. This shows that performance of MARICH decreases against DP defenses but not significantly.

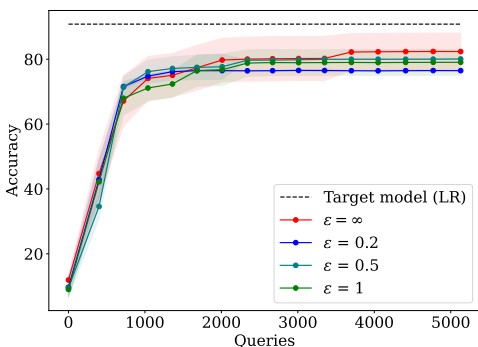

Figure 12: Performance of models extracted by MARICH against $(\varepsilon, \delta)$-differentially private target models trained using DP-SGD. We consider different privacy levels $\varepsilon$ and $\delta = 10^{-5}$. Accuracy of the extracted models decrease with increase in privacy (decrease in $\epsilon$).

**Output perturbation-based defenses.**   Perturbing output of an algorithm against certain queries with calibrated noise, in brief output perturbation, is one of the basic and oldest form of privacy-preserving mechanism [DMNS06]. Here, we specifically deploy the Laplace mechanism, where a calibrated Laplace noise is added to the output of the target model generated against some queries. The noise is sampled from a Laplace distribution $\mathrm{Lap}(0, \frac{\Delta}{\varepsilon})$, where $\Delta$ is sensitivity of the output and $\varepsilon$ is the privacy level. This mechanism ensures $\varepsilon$-DP.

We compose a Laplace mechanism to the target model while responding to MARICH's query and evaluate the change in accuracy of the extracted model as the impact of the defense mechanism. We use a logistic regression model trained on MNIST as the target model. We query it using EMNIST and CIFAR10 datasets respectively. We vary $\varepsilon$ in $\{0.25, 2, 8, \infty\}$. For each $\varepsilon$ and query dataset, we report the mean and standard deviation of accuracy of the extracted models on a test dataset. Each experiment is run 10 times.

We observe that decrease in $\varepsilon$, i.e. increase in privacy, causes decrease in accuracy of the extracted model. For EMNIST queries (Figure 13a), the degradation in accuracy is around $10\%$ for $\varepsilon = 2, 8$ but we observe a significant drop for $\varepsilon = 0.25$. For CIFAR10 queries (Figure 13b), $\varepsilon = 8$ has practically no impact on the performance of the extracted model. But for $\varepsilon = 2$ and $0.25$, the accuracy of extracted models drop down very fast.

Thus, we conclude that output perturbation defends privacy of the target model against MARICH for smaller values of $\varepsilon$. But for larger values of $\varepsilon$, the privacy-preserving mechanism might not save the target model significantly against MARICH.

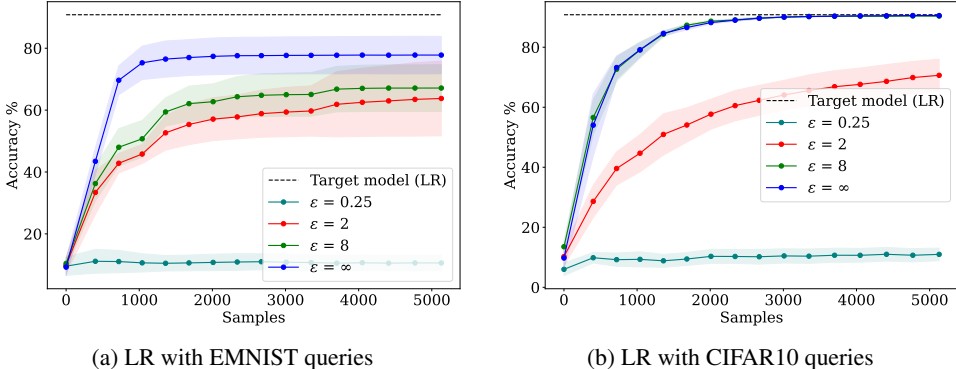

(a) LR with EMNIST queries          (b) LR with CIFAR10 queries

Figure 13: Performance of models extracted by MARICH against target models that perturbs the output of the queries to achieve $\varepsilon$-DP. We consider different privacy levels $\varepsilon$. Accuracy of the extracted models decrease with increase in privacy (decrease in $\epsilon$).

# G   Effects of model mismatch

From Equation (7), we observe that functionality of MARICH is not constrained by selecting the same model class for both the target model $f^T$ and the extracted model $f^E$. To elaborately study this aspect, in this section, we conduct experiments to show MARICH's capability to handle model mismatch and impact of model mismatch on performance of the extracted models.

Specifically, we run experiments for three cases: (1) *extracting an LR model with LR and CNN*, (2) *extracting a CNN model with LR and CNN*, and (3) *extracting a ResNet model with CNN and ResNet18*. We train the target LR and the target CNN model on MNIST dataset. We further extract these two models using EMNIST as the query datasets. We train the target ResNet model on CIFAR10 dataset and extract it using ImageNet queries. The results on number of queries and achieved accuracies are summarised in Table 6 and Figure 14.

**Observation 1.** In all the three experiments, we use MARICH without any modification for both the cases when the model classes match and mismatch. This shows *universality and model-obliviousness of* MARICH *as a model extraction attack*.

**Observation 2.** From Figure 14, we observe that model mismatch influences performance of the model extracted by MARICH. When we extract the LR target model with LR and CNN, we observe that both the extracted models achieve almost same accuracy and the extracted CNN model achieves even a bit more accuracy than the extracted LR model. In contrast, when we extract the CNN target model with LR and CNN, we observe that the extracted LR model achieves lower accuracy than the extracted CNN model. Similar observations are found for extracting the ResNet with ResNet18 and CNN, respectively.

**Conclusions.** From these observations, we conclude that MARICH can function model-obliviously. We also observe that if we use a less complex model to extract a more complex model, the accuracy drops significantly. But if we extract a low complexity model with a higher complexity one, we obtain higher accuracy instead of model mismatch. This is intuitive as the low-complexity extracted model might have lower representation capacity to mimic the non-linear decision boundary of the high-complexity model but the opposite is not true. In future, it would be interesting to delve into the learning-theoretic origins of this phenomenon.

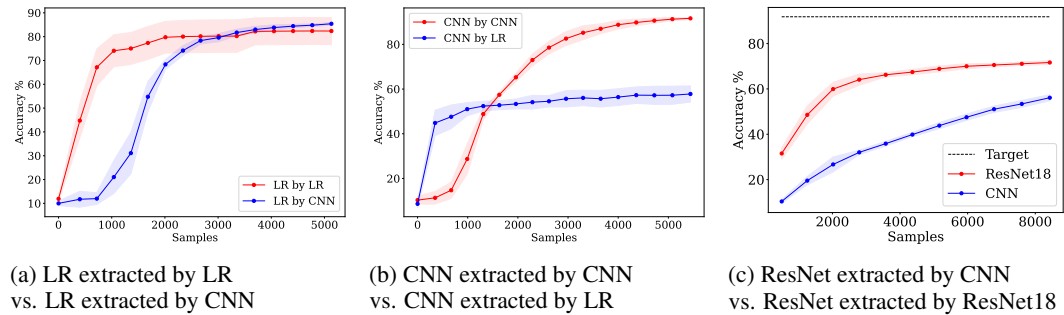

(a) LR extracted by LR vs. LR extracted by CNN

(b) CNN extracted by CNN vs. CNN extracted by LR

(c) ResNet extracted by CNN vs. ResNet extracted by ResNet18

Figure 14: Effect of model mismatches on models extracted by MARICH.

Table 6: Effect of model mismatch on accuracy of The extracted models.

| $f^E$ | $f^T$ | #**samples** | **Accuracy** |
|---|---|---|---|
| LR | LR | 5130 | $82.37 \pm 5.7\%$ |
| LR | CNN | 5130 | $85.41 \pm 0.57\%$ |
| CNN | LR | 5440 | $57.81 \pm 3.64\%$ |
| CNN | CNN | 5440 | $91.63 \pm 0.42\%$ |
| ResNet | CNN | 8429 | $56.11 \pm 1.35\%$ |
| ResNet | ResNet18 | 8429 | $71.65 \pm 0.88\%$ |

# H  Choices of hyperparameters

In this section, we list the choices of the hyperparameters of Algorithm 1 for different experiments and also explain how we select them.

Hyperparameters $\gamma_1$ and $\gamma_2$ are kept constant, i.e., 0.8, for all the experiments. These two parameters act as the budget shrinking factors.

Instead of changing these two, we change the number of points $n_0$, which are randomly selected in the beginning, and the budget $B$ for every step. We obtain the optimal hyperparameters for each experiment by performing a line search in the interval $[100, 500]$.

We further change the budget over the rounds. At time step $t$, the budget, $B_t = \alpha^t \times B_{t-1}$. The idea is to select more points as $f^E$ goes on reaching the performance of $f^T$. Here, $\alpha > 1$ and needs to be tuned. We use $\alpha = 1.02$, which is obtained through a line search in $[1.01, 1.99]$.

For number of rounds $T$, we perform a line search in $[10, 20]$.

Table 7: Hyperparameters for different datasets and target models.

| Member Dataset | Target Model | Attack Model | Attack Dataset | Budget | Initial points | $\gamma_1$ | $\gamma_2$ | Rounds | Epochs/Round | Learning Rate |
|---|---|---|---|---|---|---|---|---|---|---|
| MNIST | LR | LR | EMNIST | 250 | 300 | 0.8 | 0.8 | 10 | 10 | 0.02 |
| MNIST | LR | LR | CIFAR10 | 50 | 100 | 0.8 | 0.8 | 10 | 10 | 0.02 |
| MNIST | CNN | CNN | EMNIST | 550 | 500 | 0.8 | 0.8 | 10 | 10 | 0.015 |
| MNIST | CNN | CNN | CIFAR10 | 750 | 500 | 0.8 | 0.8 | 10 | 10 | 0.03 |
| CIFAR10 | ResNet | CNN | ImageNet | 750 | 500 | 0.8 | 0.8 | 10 | 8 | 0.2 |
| CIFAR10 | ResNet | ResNet18 | ImageNet | 750 | 500 | 0.8 | 0.8 | 10 | 8 | 0.02 |
| BBC News | BERT | BERT | AG News | 60 | 100 | 0.8 | 0.8 | 6 | 3 | $5 \times 10^{-6}$ |

