# OpenReview forum: "Marich: A Query-efficient Distributionally Equivalent Model Extraction Attack"
_NeurIPS.cc/2023/Conference — NeurIPS 2023 poster_

### Official Review · Reviewer_jbqh · 2023-07-05

**Soundness:** 3 good
**Presentation:** 3 good
**Contribution:** 3 good
**Rating:** 6
**Confidence:** 4

**Summary:**

This paper proposes a model-oblivious query-efficient black-box model extraction attack. This attack is achieved by solving a proposed distributionally equivalent and max-information model extraction problem. To solve the problem, this paper develops an active sampling-based query selection algorithm, MARICH, to select the most informative queries that simultaneously maximize the entropy and reduce the mismatch between the target and the stolen models. Experimental results validate the effectiveness and query efficiency of the proposed method.

**Strengths:**

S1: This paper proposes a novel notion called distributional equivalence extraction that can extend the existing notions of task accuracy and functionally equivalent model extractions. This paper also theoretically demonstrates that solving the developed optimization problem which correlates the distributionally equivalent with another proposed max-information extraction can simultaneously maximize the entropy and reduce the mismatch between the target and the stolen models.

S2: The experiments set up in the paper basically address general doubts that arose while reading the paper, which increases the credibility of the proposed method.

S3: The layout of this paper is elegant, and the overall structure and logical progression are clear. These advantages make this paper enjoyable for reading and easy to follow.


**Weaknesses:**

W1: The employed datasets which are used for training the target model being extracted are considered to be simple.

W2: Although this paper repeatedly foregrounds that their approach is designed to extract publically available APIs, there is no experiment that aims to extract models deployed in real-world APIs.

**Questions:**

Based on W2, I am wondering if you can provide practical evidence that can demonstrate that your proposed method can be used to extract real-world APIs (e.g., classifiers deployed by Amazon AWS, Google API, Microsoft  Azure)?

**Limitations:**

Please refer to the weakness and questions.

---

> ### Author Rebuttal · Authors · 2023-08-08
>
> We thank the reviewer for the time spent reviewing and encouraging comments about the novelty, presentation, and experimental results.
>
> **Querying industrial black-box APIs.** For genuine financial constraints, we cannot run experiments by querying the industrial APIs. Instead, we trained our own models and used them as the black-box APIs that predict for a given query datapoint. The code is at your disposal for reproducing the black-box models attacked in this work through API-like queries.
>
> **Experiments with larger datasets.** We would like to refer the reviewer to the full paper with appendix which is provided in the supplementary materials. This contains further experiments with larger datasets, such as ImageNet, being used as the public dataset for the attacks.
>
> We hope that our response addresses the reviewer's concerns. Let us know if you have any other query.

---

> > ### Comment · Reviewer_jbqh · 2023-08-16
> >
> > Thank you for your response. While I believe the assertion regarding the effectiveness of extracting real-world API might be slightly overstated, you have addressed most of my primary concerns. Therefore, I am currently inclined to maintain my current rating for this paper.

---

### Official Review · Reviewer_wp9P · 2023-07-06

**Soundness:** 3 good
**Presentation:** 2 fair
**Contribution:** 2 fair
**Rating:** 4
**Confidence:** 5

**Summary:**

Given a dataset DQ and blackbox access to a model trained on a dataset DP,  the authors identify a subset of the dataset DQ that can be used to train another model. They use a metric based on energy to select the samples.

The authors use with model stealing/extraction attacks as their primary related work, though the work has more resemblance to the literature on coresets. In coresets, DP and DQ are the same. But, overall, the algorithm MARICH seems very similar to what would be used in the coresets work. The mismatch between DP and DQ appears to be relatively low in the experiments. For example, MNIST versus EMNIST; CIFAR 10 versus CIFAR10 (same?); and BBC News versus AGNews. Notably, the DQ datasets are about the same or larger than original datasets.

It seems a better baseline for the work would be substantial work on coresets, which also aims to find an efficient subset of a dataset that can be used to train a comparable model to one that is trained on the full dataset.  The reason is that DP and DQ are not that different in authors' experiments. Indeed, if trained on 100% of the dataset, the accuracy appears to be very high. Thus, it seems an adversary has little incentive to select a subset, if the primary objective is to create a high-fidelity clone. Indeed, prior work on model stealing generally aims to exceed the performance of what would be achievable with just DQ.

Authors cite one of the coresets papers on that [lines 497-498, SS18].  But, in the last 5 years, there is a substantial body of work on coresets based on various metrics, including energy scores, forgetting, AUM, and Coverage-Centric core-set selection based on stratified sampling. I would have liked to see a comparison with that instead.





**Strengths:**

Entropy-based sampling is a reasonable strategy to reduce the size of the selected state. They use 3 datasets, MNIST, CIFAR10, and BBC News (on BERT). Use of BERT in the evaluation is good, since most other papers on coresets use


**Weaknesses:**

The paper does not appear to be well-positioned with respect to prior work.

Experiments could be better argued. Membership Inference attack is an interesting metric to use in the paper for comparing the extracted model with the original one. Is there a privacy goal here?  Why not use a straight Accuracy measure on a test dataset from the original dataset DP, as in most core-set work or model extraction work?



**Questions:**

Compare more extensively with core-set work in both related work and in the experiments.

Use metrics and datasets that are similar to those in your baseline work, so that one can more directly compare your results with those in earlier work.



**Limitations:**

I did not see a discussion on limitations in the paper or potential negative societal impact of the work.

---

> ### Author Rebuttal · Authors · 2023-08-08
>
> We thank the reviewer for his/her feedback. We answer below to his/her concerns and comments.
>
> **Explaining the experimental design:**
>
> **1. MI attack:** A membership inference attack shows informativeness of the extracted model, while never using the private training dataset. Our experimental results validate our hypothesis that since Marich is derived from the principle of max-information attack, it extracted models yielding high MI accuracy. This is a direct violation of privacy (in MI attack sense) of the users in the private training dataset.
>
> **2. Accuracy measure:** We agree with the reviewer about the test dataset and this is exactly what we have done. We use a subset of the private dataset as the test dataset for both the target and extracted models. We mention this also in our experimental goal ``How do the accuracy of the model extracted using MARICH \textbf{on the private dataset} compare with that of the target model, and RS with same query budget?" (Line 305 and 306)
>
> **Core-set literature:** Thanks for pointing us to the core-set literature. We are aware of this growing field, and we have compared our results with the models extracted using a compatible core-set approach, such as [SS17]. We have a detailed discussion on the related active sampling strategies in Appendix C of the extended paper in the supplementary material.
>
> Here, we discuss some other interesting works in this direction. We show why they are not directly implementable in our problem.
>
> [Kil+21b] aims to identify a subset of training dataset for training a specific model. The algorithm needs white-box access to the model, and also needs to do a forward pass over the whole training dataset. Thus, the objective of our work is different. Also a white-box sampling algorithm and the assumption of being able to retrieve predictions over the whole training dataset are not feasible in our problem setup.
>
> [Kil+21a] requires access to the average loss over the training dataset or a significantly diverse validation set. Then, the gradient of loss on this training or validation set is compared with that of the selected minibatch of datapoints. In a black-box attack, we do not have access to average loss over a whole training or validation dataset. Thus, it is not feasible to deploy the proposed algorithm.
>
> [MBL20] proposes an elegant preprocessing algorithm to select a coreset of a training dataset. This selection further leads to an efficient incremental gradient based training methodology. But in our case, we sequentially query the black-box model to obtain a label for a query point and use them further for training the extracted model. Thus, creating a dataset beforehand and using them further to preprocess will not lead to an adaptive attack and also will not reduce the query budget. Thus, it is out of scope of our work.
>
> [KS22] needs an auxiliary classifier to first use a subset of labelled datapoints to create low dimensional embeddings. Then  according to the query budget, it chooses the points from the sparse regions from each cluster found from the low dimensional embeddings. This is another variant of uncertainty sampling that hypothesises the points from the sparse region are more informative in terms of loss and prediction entropy. This design technique is at the same time model specific, thus incompatible to our interest, and also increases the requirement of queries to the target model.
>
> We will add this discussion in the final version of our paper and add designing core-set based algorithms as an interesting future work.
>
> **Comparison with baselines and performance metrics.** We have shown our performance on all the relevant metrics of model extraction attacks (e.g. accuracy on test dataset (App. D.1.) [MSDH19, CSBB+18, OSF19], parametric fidelity (App. D.3) and prediction matching (App D.2.2) of the extracted models [LM05, BBJP18, JCB+20]). We also add new metrics such as KL-divergence between predictive distributions of the extracted and target models as a goodness of extraction attack (Figure 3, Appendix D.2.1), and accuracy of MI attacks with extracted models as informativeness of the extracted models (Table 1, Appendix D.4).
>
> We have also compared with the previously used active learning algorithms for attacks. We would like to emphasise that this is not a work regarding generic active learning or training on a subset of dataset. The goal of this work is to design an adaptive and frugal model extraction algorithm from the first principles of distributionally equivalent/max-information extraction. Since the final algorithm is similar to an active query selection algorithm in spirit, we compare its performance with 5 other types of active selection approaches, namely  ``K-Center (KC) [SS18], Least-Confidence sampling (LC) [LS06], Margin sampling (MS) [BBZ07 , JG19], Entropy Sampling (ES) [LG94], and Random Sampling (RS)". Designing better active query selection algorithm using any of these five approaches would be an interesting research work but out of the scope of this paper.
>
> Hope we have answered your comments and concerns, and also look forward to respond if you have any further query. Hope our response will convince you to raise the score.
>
> **New references**
>
> [MBL20] Baharan Mirzasoleiman, Jeff Bilmes, and Jure Leskovec. “Coresets
> for data-efficient training of machine learning models”. ICML 2020
>
> [Kil+21a] Krishnateja Killamsetty et al. “Grad-match: Gradient matching based
> data subset selection for efficient deep model training”. ICML 2021
>
> [Kil+21b] Krishnateja Killamsetty et al. “Retrieve: Coreset selection for effi-
> cient and robust semi-supervised learning”. NeurIPS 2021
>
> [KS22] Yeachan Kim and Bonggun Shin. “In Defense of Core-set: A Density-
> aware Core-set Selection for Active Learning”. ACM SIGKDD 2022

---

> > ### Comment · Reviewer_wp9P · 2023-08-15
> >
> > Thanks for your response and clarifying some of the points.  I have some follow-up questions/concerns:
> >
> > 1. Is this understanding generally correct?  "The mismatch between DP and DQ appears to be relatively low in the experiments. For example, MNIST versus EMNIST; CIFAR 10 versus CIFAR10 (same?); and BBC News versus AGNews. Notably, the DQ datasets are about the same or larger than original datasets."
> >
> > 2. For the datasets where the public dataset is generally a superset of the private dataset (even if there is a slight mismatch in distribution),  what would be the performance of a baseline in which the adversary simply trains a model on the public dataset (no querying or active sampling needed), but uses it on the private dataset? It seems to me that could do pretty well on some measures.
> >
> > 3. Why should an MLaaS platform worry? Here is perhaps a cynical argument. If I were an MLaaS platform, it seems a competitor stealing my model using the techniques from the paper shouldn't worry me much --  with the accuracy gap between the stolen model and my model, it seems I would have no reason to worry about my market share or competitive advantage being lost. And, even if you get some idea of the distribution of the private dataset, is it accurate enough for it to really be a source of worry or competitive threat?
> >
> > 4. There is some recent work on corset selection with high pruning rates by Zhang et al. (ICLR 2023)  that would be interesting to compare with or adapt, since that is aiming for high performance with very small coresets.  Overall though, the writeup needs to draw a better contrast with the general field of coreset selection. A possible angle is to better argue that you are using the teacher (private) model to inform the coreset selection, whereas in the standard coreset world, one uses a model trained on the same dataset to inform the coreset selection.
> >
> > Thanks.

---

> > > ### Author Response · Authors · 2023-08-16
> > > **Author response to additional questions**
> > >
> > > We thank the reviewer for the response to our rebuttal. Here we address your concerns in further detail.
> > >
> > > Q1. **(a) Choice of DQ:** Only prior knowledge required to choose DQ is the data-type of DP, i.e. whether the target model uses images or texts. Under the mild assumption that the attacker knows the data-type, we conduct experiments with a significant subset of datasets used in public data-based model extraction attacks, e.g. MNIST, CIFAR10 [PMG+17, JSMA19, PGS+20], AGNews [PGS+20]. We have also included datasets, such as ImageNet, EMNIST, BBCNews, which are novel in this context.
> > >
> > > **(b) Mismatch of DP and DQ:** For Marich, the datasets DP and DQ can be significantly different. For example, we have attacked MNIST-trained model with both EMNIST and CIFAR10. Though MNIST contains handwritten digits, CIFAR10 contains images of aeroplanes, cats, dogs etc., and EMNIST contains handwritten letters. Thus, the data generating distributions and labels are significantly different.
> > >
> > > We have not attacked a CIFAR10-trained model with CIFAR10 as DQ. Rather, we have attacked a CIFAR10-trained ResNet with ImageNet as DQ. These two datasets are also known to have very different labels and images. Thus, we would respectfully disagree that in our experiments, DP and DQ are close in terms of data distributions. Rather, our experiments show that Marich can handle both data and model mismatch, which is an addendum to the existing model extraction attacks.
> > >
> > > **(c) Size of DQ:** Here, the 'size' of the dataset DQ is neither a benefit nor a loss as we aim to use small number of queries. It is the task of the attacker to reduce the number of queries for any given DQ. For example, Marich uses only 1.92% of CIFAR10 to attack an MNIST-trained logistic regression model, and 16.58% of ImageNet to attack CIFAR10-trained ResNet.
> > >
> > > Q2. **Attacks without queries:** We simulate Random Sampling (RS) (Line 300), which is the closest to the proposed approach that does not use any data-adaptive/active sampling. In RS, we uniformly sample input from DQ, send them for querying, and use the labels predicted from the target model to train the extracted model (App. C, page 18). Our experiments show that models extracted by RS are significantly low in performance w.r.t. the models extracted by Marich and other adaptive query selection algorithms (Table 2-4, Appendix D, page 26-27). This shows effectiveness of active sampling based methods than the proposed approach.
> > >
> > > But we do not understand how we can "extract a target model" without using any information from it. The definition of extraction attacks states that *"The typical setup for a model extraction attack is the one of an API, such as the ones provided by MLaaS platforms, or a model served on an edge device, such as the image models found in many of our smartphones. In both cases, the adversary is able to send inputs to the model and observe the model’s prediction, but they do not have access to the model’s internals."* [(M. Jagielski and N. Papernot)](https://cleverhans-lab.github.io/2020/05/21/model-extraction.html) Thus, even if the proposed attack without queries works under some performance measures, it would be a new type of attack, which is out of the scope of the work on extraction attacks.

---

> > > > ### Author Response · Authors · 2023-08-16
> > > > **Author response to additional questions (continued)**
> > > >
> > > > Q3. **Importance of extraction attacks:** Model extraction attack is a fertile sub-field of privacy attack design, which is concerning for following reasons.
> > > >
> > > > (a) Functionality of a model trained by an organisation is an intellectual property. *"In the security jargon, we say that this attack targets the model's confidentiality."* [(M. Jagielski and N. Papernot)](https://cleverhans-lab.github.io/2020/05/21/model-extraction.html) If one can steal a model worth millions up to 80-90% accuracy, they can use that model as their starting point and can further fine tune it as they collect more application-specific data. This is a violation of technological confidentiality, equivalent to reconstructing a proprietary machine without permission.
> > > >
> > > > (b) Another concern is privacy of the users in the private training data. *The extracted models can be used to conduct membership inference (MI) attacks on the private dataset DP, which was never touched by the adversary.* We experimentally demonstrate this issue  across models and data types (Table 1, Fig. 9-10, Table 2-4 in App. D.4.). MI attacks on the private data DP using models extracted by Marich achieve >84% accuracy, i.e. 96-100% of MI attack accuracy achieved with the target models. As model extraction can lead to almost as good MI as publicly exposing the original model, MLaaS platforms legally have to care to ensure privacy of their contributors.
> > > >
> > > > Due to these well-acknowledged concerns, extraction attack yielded interesting works from both academia and industry. We conclude with the quote that *"Model extraction is far from being a simple vulnerability. Our work shows that we still understand relatively little of the attack surface."* [(M. Jagielski and N. Papernot)](https://cleverhans-lab.github.io/2020/05/21/model-extraction.html). We believe that our paper is a humble contribution in that direction.
> > > >
> > > > Q4. **Distinction from the existing coreset algorithms:** We agree that coreset selection algorithms constitute a growing and interesting branch of active learning and subsample-based training literature. But as we have pointed out in our rebuttal that the majority of the existing works in *coreset selection algorithms use the full training dataset to select an informative subset of the training dataset, which is good to train an accurate model* (including Zheng et al, ICLR 2023).
> > > >
> > > > As you have correctly pointed out and we discussed in our rebuttal, *in model extraction, we cannot use the labels of the query dataset but we have to query a private target model as a source of labels*. Thus, the private target model can be perceived as a teacher model, whom we want to query as less as possible.
> > > >
> > > > We will use the additional page available in the final version to emphasize this difference and also to add the citations of coreset selection papers mentioned in the rebuttal. We will also mention in the future work that "As the distributionally equivalent and max-information attack principles lead to an active query selection algorithm, it would be also an interesting research direction to extend the modern active learning algorithms, such as the coreset selection algorithms, to extract target models query efficiently". We hope that this will help acknowledging the future opportunities to design coreset based algorithms for model extraction attacks.
> > > >
> > > > We thank you for the interaction. We hope to have answered your concerns. If you agree, we would appreciate if you can adjust your score accordingly. If you have further questions, please feel free to ask.

---

> > > > > ### Comment · Reviewer_wp9P · 2023-08-17
> > > > >
> > > > > I think the paper's pitch needs to be significantly revised. I think there are potential ideas here, but depending on the pitch, the paper would need to be revised a lot. For instance, if the main thrust of the paper is that distribution matching is interesting to steal because it allows membership inference attacks, you would need to more directly compare with the Shokri et al. paper in 2017 (IEEE Symp. on Security & Privacy and also on arXiv -- you cited it), possibly on their datasets or examples (they attacked commercial MaaS platforms such as google and Amazon). The overall approach seemed pretty similar, though they were focused on not "stealing" a model but on showing that potent MI attacks are feasible. For instance, they also assumed only blackbox access to a model, no access to the training dataset, and they used what they called "shadow models" that imitate the behavior of the target model.
> > > > >
> > > > > I am sure there are differences in details with Shokri et al, but the larger point is that figure out the main thrust of the paper and why the problem is important. If the main advantage you see for the problem is Membership Inference attacks, then a more direct comparison with the best work in that area is warranted -- not the work on model stealing. On the other hand, if the main objective is model stealing, then I think a smaller accuracy gap with the original model is likely needed -- given where we are in the general area of model stealing.
> > > > >
> > > > > Overall, the paper has a confusing message. With an improved pitch and figuring out the central contribution, I think it can be a good paper. But, I don't think it is quite there yet. I still think it is currently a reject. I think for the paper to have impact, it will need a significant revision, including potentially new experiments. I did update my ratings. Thanks for all the clarifications.

---

> > > > > > ### Author Response · Authors · 2023-08-19
> > > > > > **Author response to reviewer's assessment**
> > > > > >
> > > > > > Dear Reviewer,
> > > > > >
> > > > > > Thanks for recognising the merit of the ideas and contributions in this paper. Also, thanks for reconsidering the score.
> > > > > >
> > > > > > We think that there is still a little misunderstanding about our response. Our goal is not to conduct a distributional extraction to perform better MI attacks. Our hypothesis is that (a) conducting "distributionally equivalent" extraction is enough to replicate the interesting functionalities of the classifier, such as task accuracy and prediction agreement, (b) conducting "max-information" attack leads to an informative replica of the target model. A variational formulation leads to a single active learning algorithm to achieve both the goals. Since there is no direct measure of informativeness of a model, we compare the informativeness of the extracted model w.r.t. that of the target model by performing MI attacks on both. We show that the models extracted by Marich lead to similar MI attack accuracy as that of the target models (across datasets and models).
> > > > > >
> > > > > > Shokri et al. in their work propose the shadow training method of MI attack. But in the best of our knowledge, their goal is not to extract the model. Also, though often shadow models are used for designing efficient MI attacks, extracting the model is not a necessary condition for a successful MI attack. Here, we do not challenge the SOTA of MI attack. Rather, we use MI attacks to show informativeness of our model and potential privacy risks.
> > > > > >
> > > > > > Hope this clarifies the confusion.

---

> ### Comment · Area_Chair_kfnU · 2023-08-13
>
> Hi there,
>
> Thanks a lot for helping with the reviewing process at NeurIPS. Could you please interact with the authors to ensure your concerns are sufficiently addressed, or are there fundamental flaws that cannot be fixed? It would be great if you could please make your questions more concrete for the authors to respond to.
>
> Thank you.
>
> Your AC

---

### Official Review · Reviewer_tspy · 2023-07-07

**Soundness:** 3 good
**Presentation:** 3 good
**Contribution:** 3 good
**Rating:** 5
**Confidence:** 3

**Summary:**

The authors studied black-box model extraction, which is a practical scenario in MLaaS. In order to boost attack efficiency in this extraction, they focused on distributional equivalence and max-information model extraction. As for distributional equivalence, they proposed a distributional notion of equivalence. On the other hand, they maximised the mutual information between the extracted and target models’ distributions. Experiments showed the effectiveness of MARICH.

**Strengths:**

+ The topic is of practical importance.
+ The proposed two methods are novel and elegant.

**Weaknesses:**

- The writing section of this paper can be further improved and reorganized. The related work can be presented as a separate section in the paper. The results in Table 1 should be presented more clearly, especially the effectiveness of MARICH.
- The experiments may not be comprehensive enough. For the field of computer vision, only MNIST and CIFAR-10, which are relatively small datasets, were considered. Since this paper focuses on black-box model extraction in real-world scenarios, I suggest conducting experiments on ImageNet or its subsets.
- In the results of Table 1, MARICH does not show overwhelming advantages and even performs worse than entropy sampling on STL/ResNet-18. Can the tradeoff between them be analyzed?

**Questions:**

refer to weakness.

**Limitations:**

refer to weakness.

---

> ### Author Rebuttal · Authors · 2023-08-08
>
> We would like to thank the reviewer his/her valuable time spent reviewing.
>
> **Improved presentation of Table 1:** We would like to refer the reviewer to the full paper with appendix which is provided in the supplementary materials. We apologize for the fact that we had made the rectifications in Table 1 in the full version, which contains the actual representation of our experimental results.
>
> **Experiments with ImageNet:** We have actually done experiments with ImageNet/ResNet-18. We have described this in the text of the main paper (line 297, line 321-322) and later added the rectified Table 1 in the extended paper. Please check the extended paper in the supplementary material.  The results show applicability of our algorithm to larger datasets.
> We would also like to attract the reviewer's attention to the fact that the row with STL/ResNet-18 in Table 1 in the first draft is rectified in the extended version.
>
> **Trade-off of different performance measures:** We observe that even with the limited query budget (less than most of the related works), we achieve better performance in most the cases, or close enough to that of the ``Best of the Competing" attacks whenever we cannot (Table 1 in extended paper in the supplementary material). It would be interesting to check whether performance of different attacks on different datasets/models can be characterised. This can be an interesting future question and is also related to the instance hardness of privacy attacks as in (Carlini et al., 2022: https://arxiv.org/abs/2112.03570). Since the question under study is not to design a better active learning algorithm but to design an adaptive extraction attack from first information-theoretic principles, different performance measures do not only depend on the query selection algorithm and the dataset but also the information leaked by its output of a model regarding its input.
>
> We hope that our response addresses the reviewer’s questions, and will convince him/her to raise the score.

---

> > ### Comment · Reviewer_tspy · 2023-08-21
> >
> > Thank you for your response. I will consider raising my scores.

---

> > > ### Author Response · Authors · 2023-08-21
> > > **Authors' response to reviewer's official comment**
> > >
> > > Thanks for your consideration of our rebuttal. As the discussion period comes to closure, we are looking forward to your revised score.

---

### Official Review · Reviewer_SZQw · 2023-07-10

**Soundness:** 4 excellent
**Presentation:** 3 good
**Contribution:** 3 good
**Rating:** 7
**Confidence:** 2

**Summary:**

The authors propose a model extraction attack which queries a model's publicly available API and chooses samples based on maximizing entropy of the target model's predictions, and maximizing agreement with between extracted model and target model predictions. These samples are then used in training a surrogate model.

**Strengths:**

1. The paper is very well presented.
2. The authors are very clear and good about motivating their proposed method.
3. The results of the attack are quite impressive when compared to baselines.
4. Experimentation is thorough, including a range of target models and data distributions considered.

**Weaknesses:**

1. While the attack is well motivated, it seems like the major practical contribution of this work is the addition of the entropy term in eq. 7. If so, this seems like a relatively minor contribution. I don't have background in this area, but it seems like any previous work should at least consider the Model-mismatch term in your optimization problem. Is this not the case? Nonetheless, I do think the work offers a principled approach to the problem, and has valuable contributions, so this is a minor weakness in my eyes.

Update:
I have read the authors' response to my review and my concerns are largely satisfied. I will keep my score as a 7.

**Questions:**

1. Why does the definition for "Distributionally equivalent model extraction (Def. 3.1) measure the divergence between joint distributions? It seems like just considering the divergence between the distribution of the two model model outputs would be sufficient.

**Limitations:**

The authors do not include a limitations section in their main body.

---

> ### Author Rebuttal · Authors · 2023-08-08
>
> We would like to thank the reviewer for acknowledging the strengths and soundness of the contribution as well as for their comments to improving the manuscript.
>
> **Novelty of contributions:** We refer to the general comments for an in-depth discussion.
>
> **Formulation of distributionally equivalent model extraction:** A classifier induces a predictive distribution over labels/output for a given input. But it is a local view of the classifier. What we want to replicate here is the global predictive distribution of the classifier, i.e. how it induces a predictive distribution over all possible input which are sampled from a dataset or equivalently, a data-generating distribution. Later, we observe the importance of this formulation as it leads to the ERM-like query selection objective in Eq. (5) and (6) (Line 224, page 5). If we would have considered the KL divergence between only the pointwise predictive distributions, we would not have obtained the $E_Q$ term in Eq. (5) and thus, no natural ERM formulation as in Eq. (6).
>
> We hope that our response addresses the reviewer's questions. Let us know if you have any further query.

---

### Author Rebuttal · Authors · 2023-08-08

We would like to thank the reviewers for their valuable time and efforts towards improving the manuscript. In the following, we highlight the novelty of our contributions and width of our evaluation. We then address comments specific to each reviewer by responding to them directly.

**Novelty of contributions and width of evaluations:** The contributions of this work are three folds:

1. Rather than aiming for extracting replicating the task accuracy or the functional equivalence (i.e. replicating the model weights), we focus on extracting the predictive distribution of a model given a data generating distribution. We encode this objective in two ways: (a) distributionally equivalent extraction, (b) max-information extraction. Our hypothesis is that if we can extract the predictive distribution of a model that is good enough to replicate other properties of the model (e.g. accuracy) and also to run other attacks (e.g. membership inference).

2. We show that both of the problem formulations (i.e. distributionally equivalent extraction and max-information extraction) lead to a unique variational optimisation problem. This optimisation problem provides us a method to adaptively and sequentially choose queries from another dataset, without accessing any side information about or part of the private training dataset. Previously, researchers have deployed entropy sampling and other active sampling methods to efficiently select queries for attack. But our natural formulation shows that these methods can be further grounded on the objectives of distributionally equivalent extraction and max-information extraction. Thus, though the end result is an active query selection algorithm, the fundamental approach to design it is different and novel with respect to the existing works.

3. We now test our proposed algorithms on multiple types of image (MNIST, CIFAR10, EMNIST, ImageNet etc.) and text (BBCNews, AGNews) data, and different types of models (logistic regression, CNN, ResNet-18, BERT). Experiments demonstrate three things. (a) Our methodological approach to design the adaptive extraction algorithm is query efficient. (b) Our hypothesis that replicating the predictive distribution of a model is enough to replicate its functionality (e.g. accuracy) and use it for MI attacks is valid. (c) Our approach allows us to design a model-oblivious and dataset-oblivious approach to attack as we can extract the true model's predictive distribution with a different model architecture and a mismatched querying dataset.

Hope the commentary clarifies the motivation behind the problem and its relevance. We are looking forward to respond if you have further questions.

---

### Decision · Program_Chairs · 2023-09-21

**Decision:**

Accept (poster)

**Comment:**

The paper presents a novel model extraction attack utilising public APIs. It effectively motivates its approach, demonstrates impressive results against baselines, and conducts thorough experiments across various models and data distributions. However, certain weaknesses exist in terms of contribution significance, organisation, and real-world applicability.

Reviewers raised concerns about extra datasets (e.g. ImageNet) and, at times, over-claiming. Some of the concerns have been addressed in the extended version. I strongly encourage the authors to take these comments and revise the camera-ready version accordingly. In the end, the overall agreement of the reviewers was to accept the paper, and I concur.